# A Closed-Form Conic Projection for Structured Neural Network Sparsity

## Abstract

Deep neural networks require sparsity mechanisms that are both scale-invariant and computationally efficient. In this paper, we introduce a new *Cone Alignment Index* (CAI), a convex constraint whose level sets form a Lorentz hypercone. This geometric structure enables the first *Closed-Form Conic Projection* (CFCP) onto such a cone, requiring only a single support-based extrapolation step and enjoying guaranteed convergence. We derive analytical expressions for: (i) computing the active support through a provably correct thresholding rule, and (ii) performing the final projection using a closed-form support-based extrapolation coefficient. Building on these results, we propose a fast bilevel projection framework for matrix sparsity. This bilevel formulation guarantees convergence and naturally induces hardware-friendly column-wise, row-wise or diagonal-wise structured sparsity.

On Gaussian data, the proposed CFCP algorithm is up to $4.5$ times faster than the Hoyer projection and $1.7$ times faster than the GSP-Hybrid method. On uniformly distributed data, CFCP is up to 6 times faster than Hoyer and $2.2$ times faster than the GSP-Hybrid approach. Moreover, CFCP exhibits invariance with respect to the data distribution (Gaussian or uniform). At the matrix level, the proposed bilevel CFCP algorithm is on the order of 10 times faster than the GSP-Hybrid algorithm using a GPU. CFCP exhibits time invariance with respect to the constraint l. When applied to transformer attention matrices on biomedical datasets and NLP benchmarks (GLUE), our method achieves up to 90% sparsity with only a minor degradation in accuracy, consistently outperforming state-of-the-art universal diagonal Big Bird masks.

[1]Anonymous Institution, Anonymous City, Anonymous Region, Anonymous Country. Correspondence to: Anonymous Author <anon.email@domain.com>.

Preliminary work. Under review by the International Conference on Machine Learning (ICML). Do not distribute.

## 1. State of the art of neural network sparsification

Modern deep neural networks (DNNs) achieve state-of-the-art performance across a wide range of tasks due to their high representational capacity, which is typically obtained through a very large number of trainable parameters (Krizhevsky et al., 2012; He et al., 2016; Vaswani et al., 2017). However, this parameter abundance entails substantial computational and memory requirements, leading to a significant carbon footprint during both training and inference (Faiz et al., 2024).

To address these challenges, a large body of research has focused on neural network sparsification, i.e., reducing the number of nonzero weights in a model. One of the earliest and most widely adopted approaches to induce sparsity in neural networks is pruning (Alvarez & Salzmann, 2016; Han et al., 2015; Frankle & Carbin, 2019). Classical pruning methods (Sanh et al., 2020), RigL (Evci et al., 2021), and SparseGPT (Frantar & Alistarh, 2023) eliminate weights using magnitude-based or gradient-based heuristics (Hurley & Rickard, 2008). While effective in practice, these approaches yield empirical sparsity without enforcing explicit geometric constraints. More recent structured pruning methods (Xia et al., 2024; Ashkboos et al., 2024) attempt to overcome some of the inefficiencies of classical pruning techniques.

Another line of work relies on regularization-based sparsity. The Least Absolute Shrinkage and Selection Operator (LASSO) (Tibshirani, 1996; Hastie et al., 2015) promotes sparsity through an $\ell_1$-norm penalty. The $\ell_0$ norm, which directly counts the number of nonzero weights, provides exact sparsity control and is scale-invariant, but it is non-differentiable and leads to combinatorial optimization problems (Louizos et al., 2018). A key limitation of pruning-based methods, as well as $\ell_1$- and $\ell_0$-induced sparsity, is their unstructured nature, which typically produces randomly distributed zero-valued weights. Since many modern hardware architectures implement multiply–add operations as a single instruction, such irregular sparsity patterns fail to translate into practical speed-ups and are poorly suited for efficient parallel execution.

To overcome the inefficiency of unstructured sparsity, research has increasingly turned toward structured sparsity methods, which aim to remove entire groups of parameters, such as filters, channels, or neurons. Group LASSO and its variants introduce regularizers that enforce sparsity at the group level (Yuan & Lin, 2006; Kim & Xing, 2010; Scardapane et al., 2017; Yoon & Hwang, 2017; Simon et al., 2013; Wen et al., 2016; Ma et al., 2019; Alvarez & Salzmann, 2016; Ohib et al., 2022). Despite their improved hardware efficiency, these methods often suffer from significant computational overhead due to the need to solve complex Lagrangian optimization problems (Friedman et al., 2010; Mairal & Yu, 2012).

An alternative to Lagrangian regularization is constrained optimization via projection methods. $\ell_1$ projection-based approaches directly enforce sparsity by projecting weight vectors onto norm balls, typically the $\ell_1$ ball, using efficient algorithms (Duchi et al., 2008; Condat, 2016). While such methods enjoy linear-time complexity, they are not scale-invariant and do not naturally induce structured sparsity. Of particular interest is the $\ell_{1,\infty}$ projection, which promotes structured sparsity by enforcing group-wise shrinkage, for instance by setting entire columns of a weight matrix to zero. Recent works have proposed efficient $\ell_{1,\infty}$ projection algorithms based on proximal methods (Bejar et al., 2021). However, their worst-case time complexity remains $\mathcal{O}(nm \log(nm))$ for a matrix in $\mathbb{R}^{n \times m}$, which may limit their scalability to very large neural networks.

An alternative is the Hoyer score (Hoyer, 2004), which balances sparsity and scale invariance. It has been successfully applied in a variety of contexts, including blind deconvolution (Repetti et al., 2015), nonnegative least squares (Esser et al., 2013; Gillis & Glineur, 2012), neural network regularization (Yang et al., 2020; Ohib et al., 2022; Thom et al., 2015), and biomedical imaging applications (Duan et al., 2019).

Large pretrained Transformer models such as BERT (Devlin et al., 2019) and RoBERTa (Liu et al., 2020) now dominate the landscape of natural language processing. These architectures are fully dense and rely on self-attention mechanisms with quadratic complexity $\mathcal{O}(n^2)$ in the sequence length $n$. Structured sparse attention mechanisms have been explored in models such as Big Bird (Zaheer et al., 2020b;a) and Reformer, which reduce computational complexity through architectural biases.

### 1.1. Contribution and Organization of this Work

In this work, we make the following contributions:

- We introduce a new *Cone Alignment Index (CAI)*, whose level sets form a convex Lorentz cone. We distinguish two cases: points lying *outside* the cone (a non-convex set) and points lying *inside* the cone (a convex set).

- We propose a *closed-form conic projection (CFCP)* algorithm that performs a single support-based extrapolation onto the cone, and we extend it to structured matrix sparsity via a *bilevel projection*, naturally inducing structured column-wise sparsity in neural networks.

- We provide a comprehensive benchmark comparing the computational efficiency of CFCP with the original Hoyer and Newton-based projection algorithms, as well as an empirical evaluation on classification tasks using transformer architectures, comparing CFCP with the universal Big Bird masks in terms of the accuracy–sparsity trade-off.

## 2. Mathematical properties of the new Cone Alignment Index (CAI)

We use the mathematical notation file *math-commands.tex* from the textbook, Deep Learning Goodfellow et al. (2016).

### 2.1. A new Cone Alignment Index (CAI)

Let us define the *Cone Alignment Index (CAI)* constraint of a vector $\boldsymbol{x} \in \mathbb{R}^n$ as:

$$H_e(\boldsymbol{x}) = \frac{(\sum_{i=1}^n x_i)^2}{\sum_{i=1}^n x_i^2} = \frac{(\mathbf{1}^\top \boldsymbol{x})^2}{\boldsymbol{x}^\top \boldsymbol{x}} \leq l \qquad (1)$$

where $\mathbf{1}$ denotes the all-ones vector in $\mathbb{R}^n$ and $l$ is the constraint.

**Lemma 2.1.** *Geometric structure. The level sets of $H_e(\boldsymbol{x})$ define a family of* second-order surfaces

$$(\mathbf{1}^\top \boldsymbol{x})^2 = l \, \|\boldsymbol{x}\|_2^2, \qquad (2)$$

*This equation corresponds to the boundary of a revolution cone with apex at the origin and axis along the diagonal direction $\mathbf{1} = (1, 1, \ldots, 1)$ and aperture angle $\delta = \arccos(\sqrt{l/n})$. For $l \in [0, n]$, the quantity $H_e(\boldsymbol{x})$ measures how well $\boldsymbol{x}$ is aligned with this diagonal axis: $H_e(\boldsymbol{x}) = n$ if $\boldsymbol{x}$ is collinear with $\mathbf{1}$, and $H_e(\boldsymbol{x}) = 0$ if $\boldsymbol{x}$ is orthogonal to it. $H_e$ is **scale-invariant**, as a direct consequence of the definition of the CAI.*
*The level set $\mathcal{C}_l = \{x : (\mathbf{1}^\top x)^2 \leq l\|x\|_2^2\}$ is (up to a change of basis aligning the axis with $\mathbf{1}$) a second-order (Lorentz) cone and is therefore convex. Note that in contrast, the level set $\mathcal{S}_l = \{x \neq 0 : (\mathbf{1}^\top x)^2 = l\|x\|_2^2\}$ is a ruled conical surface and is not convex as a set. The level set outside the cone is non-convex.*

## 2.2. Geometric Interpretation of the CAI Projection

To compute the CAI projection, we first map the original data $y_0 \in \mathbb{R}^n$ to the non-negative orthant $y \in \mathbb{R}_+^n$ by taking $y \in \mathbb{R}_+^n$, $\quad y_i \leftarrow |y_i^0|, \; \forall i \in \{1, \ldots, n\}$. We then distinguish two cases depending on the position of $y$ with respect to the CAI cone.

**Case 1: Points satisfying the constraint.** If $y$ satisfies the constraint $H_e(y) \leq l$, then $y$ already lies outside the CAI cone and no projection is required. In this case, the point in the non-convex set remains unchanged.

**Case 2: Points violating the constraint.** If $H_e(y) > l$, then $y$ lies strictly inside the CAI cone, which is a convex set, and must be projected onto the cone boundary. Since the interior of the CAI cone is convex, the projection onto its boundary is unique (see Figure 1). Specifically, the projection of a point $y$ lying inside the cone is obtained by extrapolating along a fixed direction until reaching the boundary of the CAI cone as follows:

$$x = \lambda y + (1 - \lambda)d, \quad s.t. \quad (\mathbf{1}^\top x)^2 = l \|x\|_2^2 \quad (3)$$

where $d = (\rho, \rho, \ldots, \rho)$ lies on the cone axis. Solving this problem leads to a quadratic equation in the extrapolation parameter $\lambda$

$$a\lambda^2 + b\lambda + c = 0$$

.

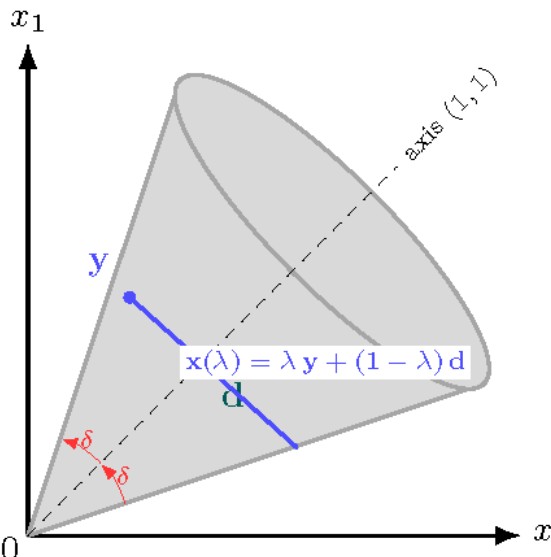

$x_1$

$y$

$x(\lambda) = \lambda\, y + (1 - \lambda)\, d$

$d$

$\delta$
$\delta$

$0$

$x_2$

axis (1,1)

*Figure 1.* Illustration of the CAI geometry

**Lemma 2.2** (Projection onto an affine Hyperplane Orthogonal to the Cone Axis). *Selecting $d$ on the cone axis and enforcing $\|d\|_1 = \|y\|_1$ guarantees that the extrapolation*

$$x(\lambda) = \lambda y + (1 - \lambda)d$$

*remains in the affine hyperplane*

$$\mathbf{1}^\top x = \|y\|_1,$$

*which is orthogonal to the cone axis. As a consequence, the extrapolation line itself is orthogonal to the cone axis, and the resulting solution $x$ satisfies the normalized $\ell_1$ constraint and thus avoids normalization or $\ell_1$ projection:*

$$\|x\|_1 = \|d\|_1 = \|y\|_1.$$

Moreover, choosing $d$ such that $\|d\|_1 = \|y\|_1$ simplifies the quadratic equation, since the linear coefficient $b$ vanishes. The coefficients reduce to

$$a = l\left(\frac{\ell_1^2}{n} - \ell_2^2\right), \quad b = 0, \quad c = \ell_1^2\left(1 - \frac{l}{n}\right). \quad (4)$$

Assume that $y$ lies *strictly inside* the cone, the projection step is a linear extrapolation beyond $y$ (along the line passing through $d$ and $y$). The closed-form value for the extrapolation coefficient $\lambda$ is:

$$\lambda = \sqrt{\frac{\ell_1^2\left(\frac{l}{n} - 1\right)}{l\left(\frac{\ell_1^2}{n} - \ell_2^2\right)}} = \sqrt{\frac{H(y)(n - l)}{l\left(n - H(y)\right)}}. \quad (5)$$

where $\ell_1$ is the norm 1 and $\ell_2$ is the norm 2 of $y$, $H(y) = \left(\|y\|_1 / \|y\|_2\right)^2$ and $(\lambda > 1)$. See proof in supplementary material.

# 3. A Closed-Form Conic Projection (CFCP) using a single Support-Based extrapolation

**Lemma 3.1** (Support-based extrapolation induces sparsity). *Let $y \in \mathbb{R}_+^n$ and let $\mathcal{C}_l$ denote the cone defined by*

$$(\mathbf{1}^\top x)^2 = l \|x\|_2^2, \quad x_i \geq 0. \quad (6)$$

*For any fixed support $S \subset \{1, \ldots, n\}$ with $|S| = \nu$, the projection of $y$ onto $\mathcal{C}_l \cap \mathbb{R}^S$ exists, is unique, and is given by*

$$x_S = \lambda\, y_S + (1 - \lambda)d_S, \quad d_S = \frac{\ell_1^{(S)}}{\nu}\mathbf{1}_S, \quad (7)$$

*where $\lambda$ is the unique positive solution of the cone equation.*

Fixing the support $S$ restricts the problem to a $\nu$-dimensional subspace. Any coordinate $y_i < \alpha(\nu)$ which lies outside the feasible cross-section must therefore be mapped to zero by the projection. Then, using Equations 5, and 6, we compute the threshold $\alpha(\nu)$ as follows:

$$\alpha = \nu^{-1}\ell_1\left(1 - \sqrt{\frac{l(\nu - H(y))}{H(y)(\nu - l)}}\right). \quad (8)$$

Therefore, given a threshold $\alpha \geq 0$, we define the hard-thresholding operator $T_\alpha : \mathbb{R}^n_+ \to \mathbb{R}^n_+$ by

$$\left(T_\alpha(\boldsymbol{x})\right)_i = \begin{cases} x_i, & \text{if } x_i \geq \alpha, \\ 0, & \text{otherwise,} \end{cases} \quad i = 1, \ldots, n. \quad (9)$$

The entries below the threshold are zeroed out.

Unfortunately, the Equation 8 can cause a division by zero problem if $\nu = l$.

**Proposition 3.2** (Support has a lower bound.). *Let $\boldsymbol{x} \in \mathbb{R}^n_+$ with support $S$ and $|S| = \nu$. By Cauchy–Schwarz applied on S,*

$$(\mathbf{1}^\top \boldsymbol{x})^2 = \left(\sum_{i \in S} x_i\right)^2 \leq \left(\sum_{i \in S} 1^2\right)\left(\sum_{i \in S} x_i^2\right) = \nu \|\boldsymbol{x}\|_2^2.$$

*Hence, dividing both sides by $\sum_{i \in S} x_i^2 = \|x\|_2^2$ yields $H_e(x) \leq \nu$.*

*If $x \in \mathbb{R}^n_+$ satisfies the cone-surface constraint, $H_e(\boldsymbol{x}) = l$. then the support size $\nu$ necessarily satisfies $l \leq \nu$.*

*Moreover, the support size $\nu \in \mathbb{N}$, thus in particular, if*

$$\boxed{l \notin \mathbb{N}, \quad then \quad \nu > l.} \quad (10)$$

*This result establishes a fundamental lower bound on the active support size and guarantees that the support-selection procedure using Equation 8 in Algorithm 1 cannot collapse below this bound.*

Based on this lemma and proposition, we propose the following algorithm: i) Ensure all components of $\boldsymbol{y}$ are non-negative. ii) Determine the active support size $\nu$ using the closed form threshold $\alpha$. iii) Compute one support-based extrapolation using the closed-form extrapolation coefficient $\lambda$, iv) restore the sign.

---

**Algorithm 1** A closed-form conic projection (CFCP) algorithm performing a single support-based extrapolation

---

**Input:** $\boldsymbol{y}, l$
$\boldsymbol{x} \leftarrow |\boldsymbol{y}_i|, \ \forall i \in [1, \ldots, n]$
$\alpha \leftarrow \frac{\|\boldsymbol{x}\|_1}{n}$
$\boldsymbol{x} \leftarrow (\boldsymbol{x}_i \text{ if } \boldsymbol{x}_i \geq \alpha \text{ else } 0 \quad \forall i \in [1, \ldots, n])$
$\nu \leftarrow \ell_0(\boldsymbol{x})$
**while** $\nu \neq \ell_0(\boldsymbol{x})$ **do**
  $\nu \leftarrow \ell_0(\boldsymbol{x})$ *(active support dimension)*
  $\alpha \leftarrow \nu^{-1}\ell_1\left(1 - \sqrt{\frac{l(\nu - H(\boldsymbol{x}))}{H(\boldsymbol{x})(\nu - l)}}\right)$
  $\boldsymbol{x} \leftarrow (\boldsymbol{x}_i \text{ if } \boldsymbol{x}_i \geq \alpha \text{ else } 0 \quad \forall i \in [1, \ldots, n])$
**end while**
$\lambda \leftarrow \frac{1}{1 - \frac{\alpha\nu}{\ell_1}}$ *(compute $\lambda$)*
$\boldsymbol{d} \leftarrow \left(\frac{\ell_1}{\nu} \text{ if } \boldsymbol{x}_j \neq 0 \text{ else } 0 \quad \forall j \in [1, \ldots, n]\right)$
$\boldsymbol{x} \leftarrow \lambda\boldsymbol{x} + (1 - \lambda)\boldsymbol{d}$
$\boldsymbol{x}_i \leftarrow \boldsymbol{x}_i \times \text{sign}(\boldsymbol{y}_i), \ \forall i \in [1, \ldots, n]$ *(restore signs)*

---

where $\lambda$ is the extrapolation coefficient, $\alpha$ is the threshold and $\nu$ is the active support size.

**Proposition 3.3** (Finite-time convergence of the active support selection ). *The iterative loop for computing $\alpha$ converges in at most $n$ iterations to a fixed point. More precisely, there exists $K \leq n$ such that:*

$$\boxed{\begin{aligned} \boldsymbol{x}^{(K+1)} &= \boldsymbol{x}^{(K)}, \\ \boldsymbol{x}^{(K)} &= T_{\alpha(\boldsymbol{x}^{(K)})}\left(\boldsymbol{x}^{(K)}\right). \end{aligned}} \quad (11)$$

See proof in supplementary material.

## 4. Relation with the Hoyer and GSP methods

The Hoyer score $H(\boldsymbol{x})$ was originally defined as the square of the ratio between $\ell_1$ and $\ell_2$ norms of the vector $\boldsymbol{x}$ (Hoyer, 2004) and update following (Yang et al., 2020):

$$H(\boldsymbol{x}) = \left(\frac{|\boldsymbol{x}|_1}{|\boldsymbol{x}|_2}\right)^2 \quad (12)$$

The GSP constraint (Group sparse Projection) following the definition (Ohib et al., 2022) is given by :

$$GSP(\boldsymbol{x}) = \left(\sum_{i=1}^{r} \frac{\sqrt{n_i} - |x_i|_1}{\sqrt{n_i} - 1}\right) \quad (13)$$

Both methods Hoyer and GSP are applied to a global non-convex set. Neither the Hoyer method nor the GSP method distinguishes between two sub-spaces. As pointed out in our CFCP method, points lying outside the cone remain unchanged, whereas points lying inside the cone (a convex set) are projected onto its boundary. The second main difference is that both the Hoyer and GSP methods use iterative algorithms, whereas our CFCP algorithm performs a single iteration.

| Property | CAI | Hoyer | GSP |
|---|---|---|---|
| Convex Cone geometry | Yes | No | No |
| Ratio norm constraint | No | Yes | Yes |
| Iterative algorithm | No (CFCP) | Yes | Yes |
| Scale invariant | Yes | Yes | Yes |

*Table 1.* Comparison between Cone Alignment Index (CAI) projection, the Hoyer projection and the GSP projection.

## 5. Bilevel CFCP Projection

Most state-of-the-art approaches for structured matrix sparsification rely on group-sparse projections and regularization techniques (Yuan & Lin, 2006; Kim & Xing, 2010; Ohib et al., 2022). Recently, bilevel optimization (BLO)

has become popular because it is a powerful method in modeling problems that involve optimizing nested objective functions (Colson et al., 2007; Zhang et al., 2022; Bennett et al., 2006; Zhang et al., 2024). BLO is an optimization problem that involves two levels of hierarchy (i.e., upper and lower levels), wherein obtaining the solution to the upper-level problem requires solving the lower-level one. In this paper, we introduce a specific bilevel $\ell_{H_e,2}$ projection that naturally induces hardware-friendly column-, row-wise or diagonal-wise (using a 45 ° rotation) structured sparsity.

Let $\boldsymbol{Y}$ be a matrix with $m$ rows and $n$ columns, and let $\boldsymbol{y}_1, \ldots, \boldsymbol{y}_n$ denote its column vectors. Let us define the row vector composed of the $\ell_2$ norms of the columns of $\boldsymbol{Y}$: $\boldsymbol{v}_2 = (\|\boldsymbol{y}_1\|_2, \ldots, \|\boldsymbol{y}_n\|_2)$, The bilevel projection optimization problem is defined as:

$$BP_l^{He,2}(\boldsymbol{Y}) = \{\boldsymbol{x} \mid \forall j,$$
$$\boldsymbol{x}_j = \arg\min_{\boldsymbol{x}} \|\boldsymbol{x} - \boldsymbol{y}_j\|_2 \quad \text{s.t.} \ \ell_2(\boldsymbol{x}_j) < u_j\}, \quad (14)$$
$$\text{where } \hat{u} \in \arg\min_u \|u - \boldsymbol{v}_2\|_2 \ \text{s.t.} \ He(u) < l.$$

A possible implementation is provided below:

---

**Algorithm 2** Bilevel $\ell_{He,2}$ Projection ($BP_\eta^{He,2}(\boldsymbol{Y})$)

---

**Input:** $\boldsymbol{Y}, \eta$
$u \leftarrow P_l^{He}(\|\boldsymbol{y}_1\|_2, \ldots, \|\boldsymbol{y}_n\|_2)$
**for** $j \in [1, \ldots, n]$ **do**
    $\boldsymbol{x}_j \leftarrow P_{u_j}^2(\boldsymbol{y}_j)$
**end for**
**Output:** $\boldsymbol{x}$

---

Where $BP_l^{He}(\boldsymbol{Y})$ is the CFCP with constraint l and $P^2$ is the classical projection on the $\ell_2$ ball. Note that the closed-form conic projection (CFCP) and the $\ell_2$ projection are closed-form algorithms, which guarantees convergence of the bilevel algorithm.

## 6. Experimental results

### 6.1. Benchmark of the Fast Closed-form cone projection versus Hoyer and GSP-Hybrid methods.

We compare our method in terms of computation time with Hoyer's projection and the Group Sparse LASSO using the GSP-Hybrid method (Ohib et al., 2022). For the implementation of the original iterative Hoyer projection, we ported the original MATLAB code of Hoyer in (Hoyer, 2004) to PyTorch. We use the efficient projection onto the $\ell_1$ ball proposed in (Duchi et al., 2008) and later corrected in (Condat, 2016). For the implementation of the GSP-Hybrid method, we use the PyTorch code in (Ohib et al., 2022).

available at [1]. Our PyTorch implementation is provided in the supplementary material.

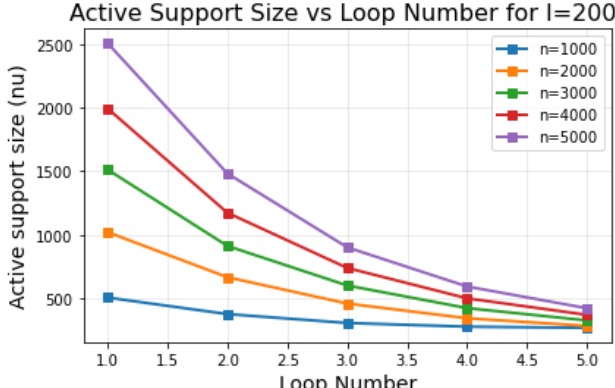

*Figure 2.* Convergence of the active support size

Figure 2 illustrates the convergence of the active support size in 5 iterations.

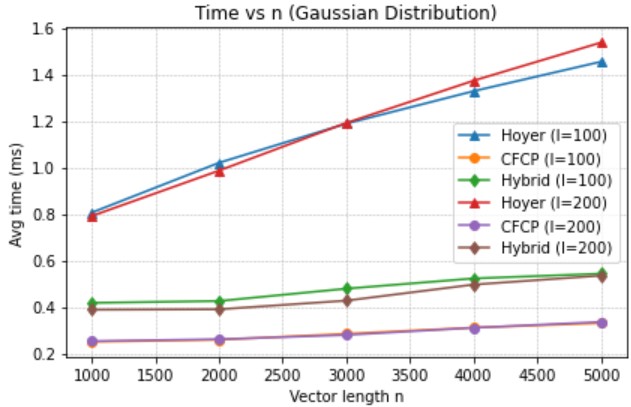

*Figure 3.* Runtime comparison of CFCP versus Hoyer, and GSP-Hybrid projection on an Apple M3 (Gaussian distribution)

Figure 3 shows that, for Gaussian-distributed data, the CFCP algorithm is approximately 4.5 times faster than the original Hoyer projection and approximately 2 times faster than the GSP-Hybrid Newton-bisection projection.

Figure 4 demonstrates that, for uniformly distributed data, the CFCP algorithm is approximately 6 times faster than the Hoyer projection and approximately 2 times faster than the GSP-Hybrid Newton-Bisection projection.

As illustrated in Table 2, the processing time is independent of the distribution for our CFCP projection, whereas it is highly dependent for the Hoyer and GSP-Hybrid projections. The standard deviation of Hoyer and GSP-Hybrid is much greater than the standard deviation of our CFCP method. Note that this discrepancy is expected, since the

---

[1] $https://github.com/riohib/gsp$

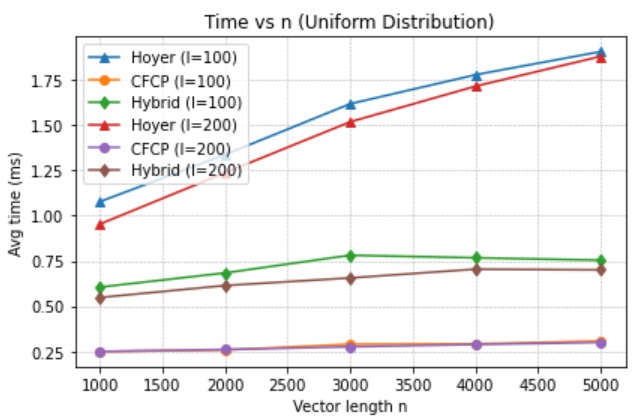

*Figure 4.* Runtime comparison of CFCP versus Hoyer and GSP-Hybrid projection on an Apple M3 (uniform distribution)

| Time | Hoyer | GSP-Hybrid | CFCP |
|---|---|---|---|
| Gaussian | 1.54 ±0.1 | 0.53 ±0.06 | 0.32 ±0.01 |
| Uniform | 1.88 ±0.08 | 0.7 ±0.04 | 0.31 ±0.01 |

*Table 2.* Time-comparison of CFCP versus Hoyer and GSP-Hybrid for $m = 5000$ and $l = 200$

original Hoyer and the GSP-Hybrid method constraints are non-convex, whereas CFCP enforces a convex constraint. Moreover, our CFCP method is invariant to data distribution (Gaussian or uniform) whereas for Hoyer and GSP-Hybrid methods, computation times increase when switching from Gaussian to uniform distributions.

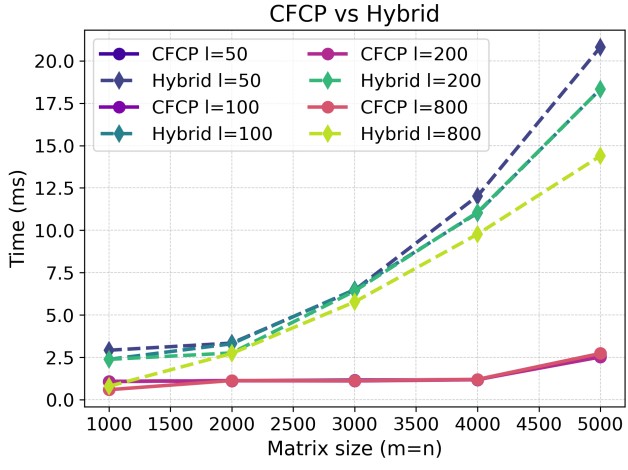

*Figure 5.* Time comparison of CFCP versus Hybrid using an NVIDIA GPU RTX 6000 Ada Generation.

Figure 5 shows that the GPU computation time on matrices of our CFCP algorithm outperforms the GSP-Hybrid algorithm in a range of 4 to 20 when the size of the attention matrix varies within a range of 1000 to 5000. Our CFCP algorithm is again independent of the constraint $l$,

| | CFCP | GSP-Hybrid |
|---|---|---|
| STD Time(ms) | 0.059 | 0.386 |

*Table 3.* Standard deviation of computation time: Comparison of CFCP versus GSP-Hybrid

whereas it is highly dependent on the GSP-GSP-Hybrid method. Table 3 Shows that the standard deviation of the computation time of the GSP-Hybrid method is 7.5 times higher than our CFCP method. Note again that this discrepancy is expected, since the original Hoyer and the GSP-Hybrid method constraints are non-convex, whereas CFCP enforces a convex constraint.

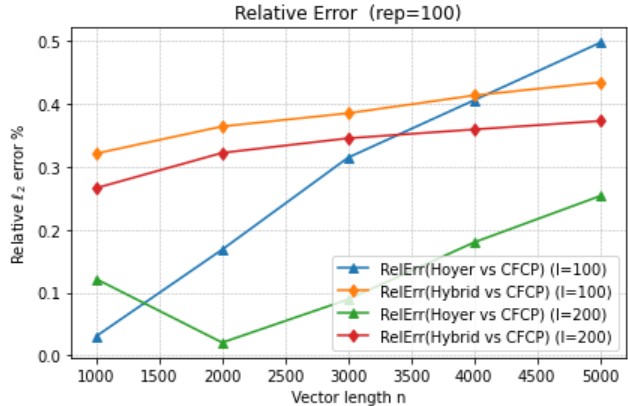

*Figure 6.* Relative norm difference between the solutions obtained by our CFCP, the Hoyer and the GSP-Hybrid projection

As illustrated in Figure 6, the relative difference between the solutions obtained with CFCP, the original Hoyer projection and the GSP-Hybrid Newton-Bisection method are similar (approximately $0.3\%$).

### 6.2. Sparsification of Attention Matrices in Transformer Architectures

**Constraint optimization**    Let $W \in \mathbb{R}^{m \times m}$ denote the attention matrix, where $m$ is the number of tokens. Let $z \in \mathbb{R}^{m \times 1}$ represent the true labels, and $z^*$ the estimated labels obtained from a softmax classifier. To sparsify the weights $W$ of the attention matrix, we employ the bilevel projection method $BP^{He,2}$ as a constraint to enforce structured sparsity in the model. The global optimization criterion is defined as:

$$\underset{W}{\text{minimize}} \quad \phi(z, z^*) \quad \text{subject to} \quad BP^{He,2}(W) \leq l, \tag{15}$$

where $\phi(z, z^*)$ is the cross-entropy loss. For minimizing this criterion, we follow the work developed by (Frankle & Carbin, 2019) and (Frankle et al., 2021) who proposed a double-descent masked-gradient algorithm. We replace their threshold with our bilevel projection.

**Application of our closed-form conic projection to the sparsification of attention matrices in transformer architectures (Vaswani et al., 2017).** Specifically, we compare our learned diagonal mask, obtained via bilevel projection, against the uniform diagonal mask of Big bird (Zaheer et al., 2020b),(Zaheer et al., 2020a). The classification framework is implemented in PyTorch, including the model, schedulers, and loss functions. We chose the ADAM optimizer (Kingma & Ba, 2015) and the smooth SiLU as activation function. For all sparsity levels and all datasets, we set the number of training epochs to 15, the batch size to 32 and the learning rate to $2 \times 10^{-5}$. We plot the curves of accuracy as a function of sparsity. The meaning of numerical parameters in the following graphs is respectively window size for Big bird and constraint level 'l' for CFCP.

**Experiment on a biomedical dataset: ECG** The challenge of the PTB Diagnostic ECG Database is formulated into a binary classification task with 10,505 abnormal and 4,045 normal ECG (Wagner et al., 2020). We report results on the Physio Net ECG dataset (Goldberger et al., 2000). The signals correspond to electrocardiogram (ECG) shapes of heartbeats which are preprocessed and segmented, with each segment corresponding to a heartbeat with 187 tokens.

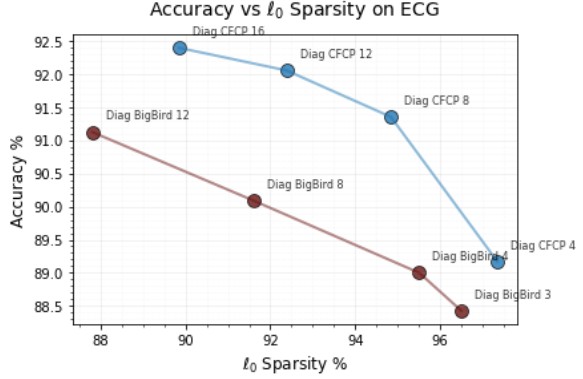

*Figure 7.* ECG dataset. Big Bird versus bilevel $\ell_{H,2}$ sparsity–accuracy trade-off.

Figure 7 illustrates that the accuracy curve as a function of sparsity. Our bilevel $\ell_{H,2}$ projection outperforms the diagonal Big bird method.

**Experiment on a natural language processing (NLP) task** while pretrained models such as BERT are fully dense Transformers, some later architectures (e.g., Big Bird, Longformer) introduce sparse attention mechanisms. Specifically, we compare these methods with our CFCP projection to a pretrained transformer-based model (Devlin

et al., 2019). Recall that a vertical band in an attention matrix corresponds to a *global token* attending to all tokens, whereas a horizontal band corresponds to all tokens attending to a global token. Such global attention patterns are known to play a critical role in Transformer-based classification models, as they enable long-range information aggregation and act as routing hubs for semantic context. From a theoretical perspective, removing these global interactions may severely limit the expressive power of the self-attention mechanism, since the first token (e.g., `[CLS]`) often serves as a summary representation of the entire sequence. Therefore, in all subsequent experiments, we explicitly include for both CFCP and Big Bird mask the first row and the first column in the diagonal sparsity mask.

**Experiments on QNLI, QQP and MLNI dataset** **The Stanford Question Answering Dataset** is a question-answering dataset consisting of question-paragraph pairs, where one of the sentences in the paragraph (drawn from Wikipedia) contains the answer to the corresponding question (written by an annotator) (Rajpurkar et al., 2016). The task is to determine whether the context sentence contains the answer to the question.

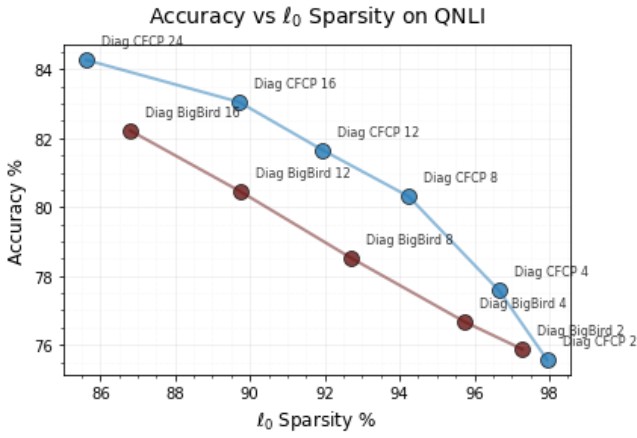

*Figure 8.* QNLI dataset. Bilevel $\ell_{H,2}$: sparsity–accuracy trade-off.

**The Quora Question Pairs2 dataset (QQP)** is a collection of question pairs from the community question-answering website Quora. The task is to determine whether a pair of questions are semantically equivalent.
For these thee datasets, we used attention matrices with 256 tokens instead of the standard 512 tokens in order to increase the difficulty of the sparsification task.

Figures 8 and 9 demonstrate that, using that global attention pathways, our CFCP method consistently outperforms the universal *Big Bird* masking strategy in terms of the accuracy–sparsity trade-off.

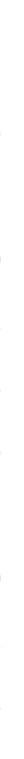

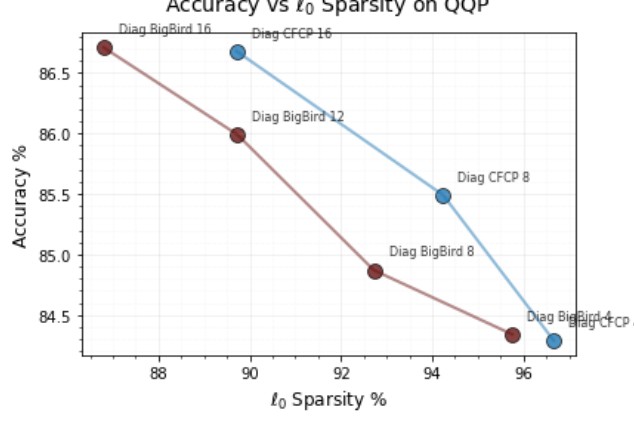

*Figure 9.* QQP dataset.Bilevel $\ell_{H,2}$: sparsity–accuracy trade-off.

**The Multi-Genre Natural Language Inference Corpus** (Williams et al., 2018) is a crowdsourced collection of sentence pairs with textual entailment annotations. Given a premise sentence and a hypothesis sentence, the task is to predict whether the premise entails the hypothesis (entailment), contradicts the hypothesis (contradiction), or neither (neutral). We report accuracy between true private labels and predicted labels.

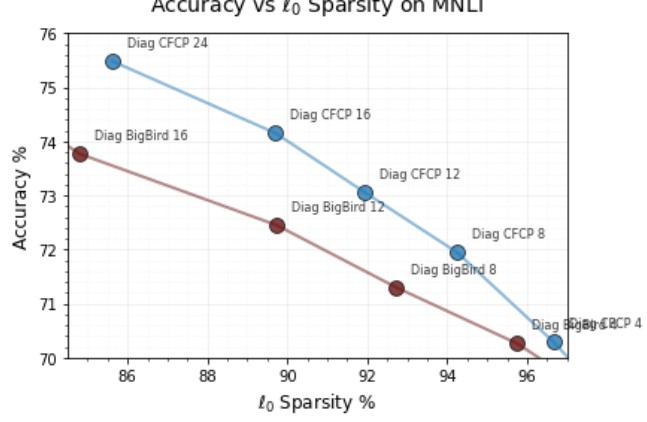

*Figure 10.* MNLI dataset. Bigbird versus bilevel $\ell_{H,2}$: sparsity–accuracy trade-off.

Figure 10 illustrates that the accuracy curve as a function of sparsity of CFCP outperforms Big Bird.

**Ablation study**  To better understand the impact of the closed-form projection, we conduct a comprehensive ablation study. We consider the following variants of our method: i) Full model (Ours): closed-form cone projection with support selection and bilevel $\ell_{H,2}$ structure. ii) No structured sparsity: standard dense model without projection. iii) Big bird projection All variants are evaluated under identical training protocols, initialization, and hyperparameters.

| Accuracy | Baseline | Diag Bigbird | Diag $\ell_{H,2}$ |
|----------|----------|--------------|-------------------|
| ECG | $89 \pm 1.9$ | $90.5 \pm 2.02$ | $92.5 \pm 1.59$ |
| QNLI | $88.52 \pm 1.2$ | $80.5 \pm 2.35$ | $84 \pm 2.4$ |
| QQP | $88.65 \pm 1.$ | $85 \pm 1.05$ | $86 \pm 1$ |
| MNLI | $78.9 \pm 1.3$ | $72.5 \pm 2.29$ | $74.2 \pm 2.3$ |

*Table 4.* Ablation. Accuracy for 90% sparsity. Baseline (no projection), Big bird, and bilevel CFCP

Table 4 reports the impact of each ablation setting on the sparsity level and predictive accuracy. Notably, on the ECG dataset, enforcing 90% sparsity leads to an improvement in accuracy, as it effectively removes noisy tokens.

We also note that using attention matrices with 256 tokens instead of the standard 512 results in a baseline accuracy approximately 3% lower than the values reported in the literature. Furthermore, for the QQP and MNLI datasets, only 30% of the data were used in order to have approximately the same number of samples in the 3 data sets. As shown in Table 4, our method achieves up to 90% sparsity with only a limited performance degradation (at most 4.5%) compared to the baseline model using full attention on NLP datasets. Moreover, for a fixed sparsity level of 90%, the learned mask obtained with our bilevel CFCP method consistently outperforms the diagonal Big Bird masking strategy. In particular, it provides an accuracy improvement of approximately 3% on the QNLI dataset, 1% on the QQP dataset, and 2% on the MNLI dataset.

## 7. Conclusion

In this paper, we introduce a new *Cone Alignment Index* (CAI), a convex constraint whose level sets form a Lorentz hypercone. This geometric structure enables the first *Closed-Form Conic Projection* (CFCP) onto such a cone, requiring only a single support-based extrapolation step and enjoying guaranteed convergence. Building on these results, we propose a fast bilevel projection framework for matrix sparsity. This bilevel projection guarantees convergence and naturally induces hardware-friendly column-wise, row-wise or diagonal-wise structured sparsity.

Our method achieves up to 90% attention sparsity with low accuracy loss in NLP GLUE datasets and outperforming state-of-the art "universal" diagonal Big Bird masks.

Moreover, our approach shows potential in broader applications, such as non-negative matrix factorization and deconvolution problems.

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

## A. Supplementary material

### A.1. Convexity of the cone and non-convexity of its exterior

**Lemma A.1** (Convexity of the cone and non-convexity of its exterior). *Let $\mathcal{C} \subset \mathbb{R}^n$ be a nonempty closed convex cone, and assume $\mathcal{C} \neq \mathbb{R}^n$. Then:*

1. *$\mathcal{C}$ is convex.*

2. *Its complement $\mathbb{R}^n \setminus \mathcal{C}$ is not convex.*

*In particular, for the CAI/Lorentz-type cone*

$$\mathcal{C}_l = \{\boldsymbol{x} \in \mathbb{R}^n : (\mathbf{1}^\top \boldsymbol{x})^2 \geq l\|\boldsymbol{x}\|_2^2\}, \qquad 1 < l < n,$$

*the feasible set $\mathcal{C}_l$ is convex, whereas the infeasible region $\mathbb{R}^n \setminus \mathcal{C}_l$ is non-convex.*

*Proof.* (1) Convexity of $\mathcal{C}$ holds by definition of a convex cone.

(2) Since $\mathcal{C} \neq \mathbb{R}^n$, there exists $\boldsymbol{u} \notin \mathcal{C}$. Because $\mathcal{C}$ is a cone, $0 \in \mathcal{C}$ and for any $\boldsymbol{x} \in \mathcal{C}$ we also have $t\boldsymbol{x} \in \mathcal{C}$ for all $t \geq 0$. Moreover, because $\mathcal{C}$ is a proper cone, there exists $\boldsymbol{w} \in \text{int}(\mathcal{C})$ (e.g., for $\mathcal{C}_l$ take $\boldsymbol{w} = \mathbf{1}$). Consider the two points

$$\boldsymbol{x}_1 = \boldsymbol{u}, \qquad \boldsymbol{x}_2 = \boldsymbol{u} + T\boldsymbol{w},$$

with $T > 0$ large. Since $\boldsymbol{w} \in \text{int}(\mathcal{C})$ and $\mathcal{C}$ is closed and convex, the ray $\{\boldsymbol{u} + t\boldsymbol{w} : t \geq 0\}$ must intersect $\mathcal{C}$ for some $t = t_0 > 0$; hence for sufficiently large $T$, $\boldsymbol{x}_2$ lies in $\mathcal{C}$ or at least the segment $[\boldsymbol{x}_1, \boldsymbol{x}_2]$ intersects $\mathcal{C}$. In either case, we can choose $T$ such that $\boldsymbol{x}_1, \boldsymbol{x}_2 \notin \mathcal{C}$ but there exists $\theta \in (0, 1)$ with

$$\theta \boldsymbol{x}_1 + (1 - \theta)\boldsymbol{x}_2 \in \mathcal{C}.$$

Therefore, $\mathbb{R}^n \setminus \mathcal{C}$ cannot be convex. $\qquad\square$

### A.2. Analysis of the Quadratic Equation

We consider the following parametric line:

$$\lambda \boldsymbol{y} + (1 - \lambda)\boldsymbol{d},$$

with $\boldsymbol{d} = (\rho, \rho, \ldots, \rho)$, and the CAI $\mathcal{H}_{e,l}$ defined by

$$H_e(\boldsymbol{x}) = \frac{(\mathbf{1}^\top \boldsymbol{x})^2}{\|\boldsymbol{x}\|_2^2} \leq l.$$

Solving for the intersection, we obtain:

$$H_e(\lambda \boldsymbol{y} + (1 - \lambda)\boldsymbol{d}) = l$$

$$\Leftrightarrow \frac{\left(\sum_i \lambda(y_i - \rho) + \rho\right)^2}{\sum_i \left(\lambda(y_i - \rho) + \rho\right)^2} = l \qquad (16)$$

After simplification, this leads to the following quadratic equation in $\lambda$:

$$a\lambda^2 + b\lambda + c = 0, \tag{17}$$

where the coefficients are given by:

$$
\boxed{
\begin{aligned}
a &= \ell_1^2 - l\,\ell_2^2 + (n-l)\big(n\rho^2 - 2\rho\,\ell_1\big), \\
b &= 2(n-l)\rho\,(\ell_1 - n\rho), \\
c &= (n-l)n\rho^2.
\end{aligned}
}
\tag{18}
$$

**Condition on $d$**   using this parameter $\lambda$ with the points $y$ and $d$ provides the following condition for ensuring a positive solution:

$$
\begin{aligned}
\rho &> n^{-1}\left(\ell_1 - \sqrt{\tfrac{l(n\ell_2^2 - l\ell_1^2)}{n-l}}\right) \\
&\Leftrightarrow \quad \|d\|_1 > \|y\|_1 - \sqrt{\tfrac{l(n\|y\|_2^2 - l\|y\|_1^2)}{n-l}}.
\end{aligned}
\tag{19}
$$

**Special case $b = 0$ (choosing $\|d\|_1 = \|y\|_1$).**   Note that when $\ell_1 = n\rho$, i.e., when $y$ lies exactly on the cone axis, the linear term $b$ vanishes and the quadratic reduces to a simpler form.

If $\ell_1 = n\rho$ (i.e., $b = 0$), the quadratic reduces to $a\lambda^2 + c = 0$ with

$$a = \ell_1^2 - l\,\ell_2^2, \qquad c = n\rho^2(n-l) = \frac{\ell_1^2}{n}(n-l).$$

solving for $\lambda > 0$ gives

$$
\boxed{
\lambda = \sqrt{\frac{c}{-a}} = \sqrt{\frac{\frac{\ell_1^2}{n}(n-l)}{l\,\ell_2^2 - \ell_1^2}} = \sqrt{\frac{H(y)\,(n-l)}{l\,(n - H(y))}}
},
$$

where $H(y) = \big(\|y\|_1/\|y\|_2\big)^2$. This is the closed form used in the CFCP projection when $\|d\|_1 = \|y\|_1$.

**Proposition A.2** ($\lambda > 1$ when $\|d\|_1 = \|y\|_1$). *Let $y \in \mathbb{R}^n_+$, $1 < l < n$, and let $d = \rho\mathbf{1}$ with $\rho > 0$ chosen such that $\|d\|_1 = \|y\|_1$ (equivalently, $\rho = \|y\|_1/n$). Consider the line*

$$x(\lambda) = \lambda y + (1-\lambda)d.$$

*Assume that $y$ violates the cone-surface constraint, i.e.,*

$$H(y)\,\frac{\|y\|_1^2}{\|y\|_2^2} > l.$$

*Then the (unique) intersection $x(\lambda) \in \partial\mathcal{C}_l$ satisfies $\lambda > 1$.*

*Proof.* Since $\|d\|_1 = \|y\|_1$, we have $\mathbf{1}^\top d = \mathbf{1}^\top y = \|y\|_1$. Hence, for any $\lambda$,

$$\mathbf{1}^\top x(\lambda) = \lambda\,\mathbf{1}^\top y + (1-\lambda)\,\mathbf{1}^\top d = \|y\|_1,$$

so the numerator $(\mathbf{1}^\top x(\lambda))^2$ is constant along the line. Therefore, enforcing $x(\lambda)$ to lie on the cone surface

$$(\mathbf{1}^\top x)^2 = l\|x\|_2^2$$

is equivalent to requiring

$$\|x(\lambda)\|_2^2 = \frac{\|y\|_1^2}{l}.$$

Now, the assumption $H(y) > l$ implies

$$\|y\|_2^2 < \frac{\|y\|_1^2}{l},$$

i.e., the target norm on the cone surface is strictly larger than $\|y\|_2$. Since $x(1) = y$, the equality $\|x(\lambda)\|_2^2 = \|y\|_1^2/l$ cannot hold at $\lambda = 1$, and the solution must occur for $\lambda > 1$ (i.e., beyond $y$ on the line). $\square$

In other words, because $\|x(\lambda)\|_1$ is fixed along the line while the cone boundary requires a larger $\ell_2$ norm than $y$ has, reaching the boundary necessarily requires extrapolation ($\lambda > 1$).

### A.3. Support-Based extrapolation

**Lemma A.3** (Support-Based extrapolation). *Let $y \in \mathbb{R}^n_+$ and let $\mathcal{C}_l$ denote the cone defined by*

$$(\mathbf{1}^\top x)^2 = l\,\|x\|_2^2, \qquad x_i \geq 0. \tag{20}$$

*For any fixed support $S \subset \{1,\ldots,n\}$ with $|S| = \nu$, the projection of $y$ onto $\mathcal{C}_l \cap \mathbb{R}^S$ exists, is unique, and is given by*

$$x_S = \lambda\,y_S + (1-\lambda)d_S, \qquad d_S = \frac{\ell_1^{(S)}}{\nu}\mathbf{1}_S, \tag{21}$$

*where $\lambda$ is the unique positive solution of the cone equation.*

*Proof.* Fixing the support $S$ restricts the problem to a $\nu$-dimensional subspace, in which the cone $\mathcal{C}_l$ reduces to a Lorentz cone with apex at the origin and axis aligned with $\mathbf{1}_S$. Within this convex subspace, the projection onto the cone is known to be unique and to lie in the two-dimensional plane spanned by $y_S$ and $\mathbf{1}_S$. Solving the cone equation within this plane yields the closed-form extrapolation expression for $x_S$. Any coordinate $y_i < \alpha(\nu)$ which lies outside the feasible cross-section must therefore be mapped to zero by the projection. Consequently, including such an index in the support contradicts feasibility. This establishes that the optimal support of the projection consists of the $\nu$ largest components of $y$. Once $\nu$ is determined, the projection is obtained via a single closed-form extrapolation step. $\square$

## A.4. Convergence of the support Thresholding Algorithm

Let $\boldsymbol{y} \in \mathbb{R}_+^n$ be a given nonnegative vector (e.g., $|\boldsymbol{y}|$ in our algorithm), and let $\boldsymbol{x} \in \mathbb{R}_+^n$ be a candidate solution. We denote by

$$\nu(\boldsymbol{x}) \;=\; \ell_0(\boldsymbol{x}) = |\{i : x_i \neq 0\}| \quad \text{and} \quad \ell_1(\boldsymbol{x}) = \|\boldsymbol{x}\|_1.$$

We also denote by $H(\boldsymbol{x})$ a sparsity score depending only on the nonzero components of $\boldsymbol{x}$ (e.g., the Hoyer or Cone Alignment Index (CAI)). For a given level $l$ and integer $\nu$, we define the threshold

$$\alpha(\boldsymbol{x}) \;=\; \frac{1}{\nu(\boldsymbol{x})}\, \ell_1(\boldsymbol{x})\left(1 - \sqrt{\frac{l\big(\nu(\boldsymbol{x}) - H(\boldsymbol{x})\big)}{H(\boldsymbol{x})\,\big(\nu(\boldsymbol{x}) - l\big)}}\right), \quad (22)$$

Given a threshold $\alpha \geq 0$, we define the hard-thresholding operator $T_\alpha : \mathbb{R}_+^n \to \mathbb{R}_+^n$ by

$$\big(T_\alpha(\boldsymbol{x})\big)_i = \begin{cases} x_i, & \text{if } x_i \geq \alpha, \\ 0, & \text{otherwise}, \end{cases} \quad i = 1, \ldots, n. \quad (23)$$

The fixed-point equation

$$\boldsymbol{x} = T_{\alpha(\boldsymbol{x})}(\boldsymbol{x}) \quad (24)$$

captures the idea that the support of $\boldsymbol{x}$ and the threshold $\alpha(\boldsymbol{x})$ must be mutually consistent: the entries below the threshold are zeroed out, and the threshold itself is computed from the nonzero entries only.

We now consider the iterative thresholding scheme used in our CFCP algorithm. Starting from $\boldsymbol{x}^{(0)} = |\boldsymbol{y}|$, we define the sequence

$$\nu^{(k)} = \ell_0\big(\boldsymbol{x}^{(k)}\big),$$

$$\alpha^{(k)} = \frac{1}{\nu^{(k)}}\, \ell_1\big(\boldsymbol{x}^{(k)}\big)\left(1 - \sqrt{\frac{l\big(\nu^{(k)} - H(\boldsymbol{x}^{(k)})\big)}{H(\boldsymbol{x}^{(k)})\,\big(\nu^{(k)} - l\big)}}\right),$$

$$\boldsymbol{x}^{(k+1)} = T_{\alpha^{(k)}}\big(\boldsymbol{x}^{(k)}\big),$$

$$\quad (25)$$

and stop as soon as the support stabilizes, i.e.,

$$\ell_0\big(\boldsymbol{x}^{(k+1)}\big) = \ell_0\big(\boldsymbol{x}^{(k)}\big).$$

**Lemma A.4** (Monotone support decrease). *For the sequence defined in equation 25, the support sizes satisfy*

$$\nu^{(k+1)} \;\leq\; \nu^{(k)} \quad \text{for all } k,$$

*and $\nu^{(k+1)} < \nu^{(k)}$ whenever $\boldsymbol{x}^{(k+1)} \neq \boldsymbol{x}^{(k)}$.*

*Proof.* By definition of $T_{\alpha^{(k)}}$, the transition from $\boldsymbol{x}^{(k)}$ to $\boldsymbol{x}^{(k+1)}$ can only set some coordinates of $\boldsymbol{x}^{(k)}$ to zero; it never activates new coordinates. Therefore, the number of nonzero entries cannot increase, i.e., $\nu^{(k+1)} \leq \nu^{(k)}$. Moreover, if $\boldsymbol{x}^{(k+1)} \neq \boldsymbol{x}^{(k)}$, at least one coordinate that was previously nonzero is set to zero, hence $\nu^{(k+1)} < \nu^{(k)}$. $\square$

**Proposition A.5** (Finite-time convergence). *The iterative scheme equation 25 converges in at most $n$ iterations to a fixed point of equation 24. More precisely, there exists $K \leq n$ such that*

$$\boldsymbol{x}^{(K+1)} = \boldsymbol{x}^{(K)},$$

*and $\boldsymbol{x}^{(K)}$ satisfies $\boldsymbol{x}^{(K)} = T_{\alpha(\boldsymbol{x}^{(K)})}\big(\boldsymbol{x}^{(K)}\big)$.*

*Proof.* By Lemma A.4, the sequence $\{\nu^{(k)}\}$ is nonincreasing and takes values in $\{0, 1, \ldots, n\}$. Therefore, it must stabilize in at most $n$ steps: there exists $K \leq n$ such that,

$$\nu^{(K+1)} = \nu^{(K)}.$$

By definition of $\boldsymbol{x}^{(K+1)}$, we have $\boldsymbol{x}^{(K+1)} = T_{\alpha^{(K)}}(\boldsymbol{x}^{(K)})$. If the support size is unchanged, then no new zero has been introduced, hence the thresholding operator leaves all nonzero coordinates unchanged. Consequently $\boldsymbol{x}^{(K+1)} = \boldsymbol{x}^{(K)}$, and $\boldsymbol{x}^{(K)}$ is a fixed point of the map $\boldsymbol{x} \mapsto T_{\alpha(\boldsymbol{x})}(\boldsymbol{x})$, which is exactly equation 24. $\square$

### A.5. Learned mask

The figures 11, 12, 13 and 14 show the masks calculated with CFCP. Unlike bib bird masks, which are uniform bands, these masks select a set of specific diagonals from each database.

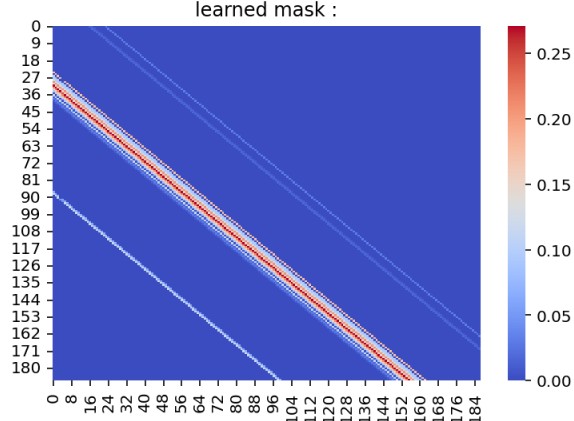

*Figure 11.* ECG dataset learned Mask.

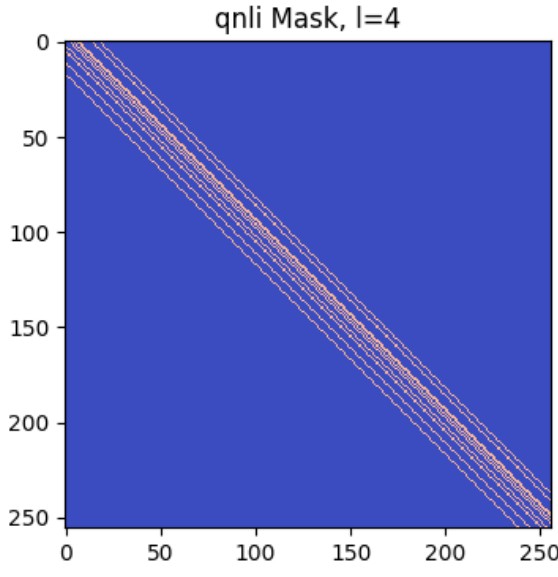

*Figure 12.* QNLI dataset learned Mask.

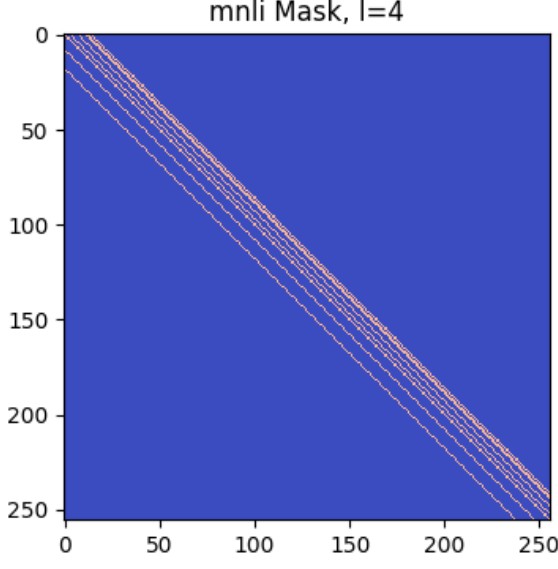

*Figure 14.* MNLI dataset learned Mask.

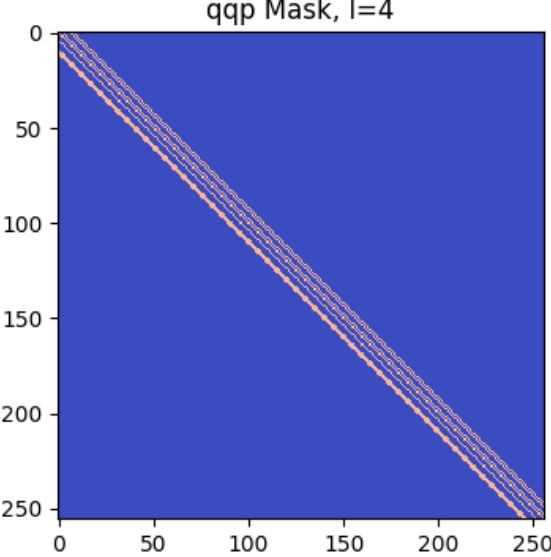

*Figure 13.* QQP dataset learned Mask.

