# OpenReview forum: "A Closed-Form Conic Projection for Structured Neural Network Sparsity"
_ICML.cc/2026/Conference — Submitted to ICML 2026_

### Official Review · Reviewer_mAH1 · 2026-03-04

**Soundness:** 3
**Presentation:** 2
**Significance:** 3
**Originality:** 3
**Overall Recommendation:** 4
**Confidence:** 1

**Summary:**

This paper introduces a new cone alignment index (CAI). CAI is a convex constraint, whose level sets form a Lorentz hyperplane. Its value is minimized when a vector is orthogonal to the vector of ones, while it is maximized when a vector is colinear with the vector of ones.

Based on CAI, a point $\mathbf{y} \in \mathbb{R}^{n}_{+}$, which violates the constraint $H_{e}(\mathbf{y}) \le l$ is projected onto the cone boundary via extrapolation. The extrapolation parameter can be computed in a closed-form solution.

Authors propose a closed-form conic projection (CFCP) scheme that zero-outs certain basis and performs extrapolation : (i) takes element-wise absolute, (ii) determines the active support size and threshold $\alpha$, zero-out the component $y_j$ if $y_j < \alpha$, (iii) performs projection onto the cone boundary via extrapolation, and (iv) restore the sign.

Finally, the authors expand the CFCP to structured sparse projection. For a matrix $Y \in \mathbb{R}^{m\times n}$, authors define a row vector $\mathbf{v}_2$ composed of the  $\ell_2$ norms of the columns of $Y$. They first perform CFCP to the vector $\mathbf{v}_2$. Subsequently, for the $j$-th column of $Y$, $\mathbf{y}_j$, they perform the classical projection on the $\ell_2$ ball, where the radius is determined by the $j$-th component of $\mathbf{v}_2$.

The proposed method demonstrates improved computational efficiency over the original Hoyer and Newton-based projection algorithms.

**Compliance With Llm Reviewing Policy:**

Affirmed.

**Final Justification:**

Thanks to authors for addressing my questions. My concerns are addressed, and I have increased my scores (3 -> 4).

**Key Questions For Authors:**

Q1: Why is the extraploation in Eq.~(3) is called projection? Is $\mathbf{x}$ is the point on the cone surface that is the closest to $\mathbf{y}$?

Q2: The while loop in algorithm 1 seems to have no effect, since we set $\nu=\ell_0(\mathbf{x})$ before the loop.

Q3: Why the coordinates $y_i < \alpha(\nu)$ are set to zero?

**Limitations:**

Yes

**Strengths And Weaknesses:**

1. Soundness:
(i) In Algorithm 1, $\nu$ is set to $\ell_0(\mathbf{x})$ before the while loop, thus the while loop has no effect.
(i) Through the line 158--160 in page 3, reasoning for zeroing of the coordinate $y_i<\alpha$ seems insufficient.

2. Presentation:
(i) Through the line 158--161 in page 3, the threshold parameter $\alpha$ suddenly appears, and described later. It will be better to describe the rationale behind thresholding first, defining the $\alpha$ second, and then explaining the zeroing procedure.

3. Significance:
Yes, this work is significant.

4. Originality:
Yes, this work provides a novel method.

---

> ### Author Rebuttal · Authors · 2026-03-28
>
> Response to Reviewer mAH1
>
> We thank the reviewer for their careful reading and constructive feedback, as well as for recognizing the originality of our mathematically grounded approach.
>
> We would like to clarify that the main contribution of this paper is a novel projection method supported by two lemmas and two theorems.
>
> Our approach combines an **exponentially convergent active set selection procedure** with a **closed-form linear extrapolation projection in a conex cone**, resulting in both theoretical guarantees and practical efficiency.
> We  provided all the PyTorch code needed to reproduce the benchmarks and experiments
>
>
> Response to  remark 1 and 2
>
> The classical  approach computes the projection and then identifies the active set (sparsity induction) corresponds to the iterative heuristic original method proposed by Hoyer.
>
> In contrast, our method first identifies the active set (sparsity induction) and then computes the projection onto the cone in a single step. Our approach is supported by two lemmas and two propositions, along with their proofs.
> The while loop (lines 209-213) is  an efficient **exponentially convergent active set selection procedure** (see Figure 2)
>
> We agree to describe the motivation for thresholding first, then explaining the zeroing procedure.
>
> As shown in Figure 5, our method achieves significant computational gains, being up to $20\times$ faster than the GSP method for $5{,}000 \times 5{,}000$ matrices and up to $80\times$ faster for $10{,}000 \times 10{,}000$ matrices  running the pytorch benchmark. We would like to thank Ohib et al. for providing us with their PyTorch code, which enabled us to perform a fair benchmark comparison. Thus we will, provide additional accuracy-sparsity results with GSP.
>
> We  provided all the PyTorch code needed to reproduce the benchmarks and experiments.
>
>
> Q1  Yes x is the point on the cone surface that is the closest to y. We adopt terminology consistent with prior work by  Hoyer and GSP.
>
> Q2: During the while loop, only the active set is updated. There is no projection using extrapolation during the while loop
>
> Q3: Fixing the support $S$ restricts the problem to a $nu$-dimensional subspace. Any coordinate $y_i < \alpha(nu)$ that lies outside the feasible region is therefore mapped to zero by the projection.

---

> > ### Author Rebuttal · Reviewer_mAH1 · 2026-04-03
> >
> > Thank you for the authors’ rebuttal. It resolved some of my earlier concerns, but one issue remains unclear.
> >
> > Q2)
> > In algorithm 1, $\nu$ is initialized as $\ell_0(x)$ before entering the while loop.
> > If $\nu = \ell_0(x)$, then the loop condition appears to be violated from the start, so the while loop would never execute.
> > This makes the algorithm difficult to follow, and I am currently unable to understand how the proposed procedure is inteded to work.
> >
> > Could the authors clarify whether there is a typo in the initiailzation or in the loop condition?

---

> > > ### Author Response · Authors · 2026-04-03
> > >
> > > In the code def cfcp provided in supplementaty material we have
> > >
> > >   line  380  nu_prev = int((x > 0).sum().item())
> > >
> > >   line   381 nu = nu_prev + 1
> > >
> > > Thus there is a typo
> > >
> > > Line 208 should be
> > >
> > > $\nu =\ell_0(x)  +1$ and not $\nu =\ell_0(x) $
> > >
> > > Thanks very much for pointing this typo

---

### Official Review · Reviewer_CtkH · 2026-03-11

**Soundness:** 3
**Presentation:** 4
**Significance:** 3
**Originality:** 3
**Overall Recommendation:** 4
**Confidence:** 1

**Summary:**

The authors introduce a conic constraint for sparsifying the weights in modern architectures. The procedure is based on projections onto a cone, which can be done in closed form, rendering the procedure extremely efficient. This procedure is compared with a number of sparsification approaches. The proposed method achieves better performance at fixed sparsity, while being fast to run.

**Compliance With Llm Reviewing Policy:**

Affirmed.

**Final Justification:**

I don't think my expertise allows me to judge the significance and the implications of this work beyond its mathematical correctness. I thus confirm my positive rating, while keeping the lowest possible confidence

**Key Questions For Authors:**

1. Have you compared the performance against other approaches like "Reformer" and Longformer"?
2. In figures 3, 4 I do not see the CFCP (l=100) line. How does it fare with respect to l=200?
3. In many points in the paper you claim that the sparsification is "hardware-friendly". What do you mean with this?

**Limitations:**

Yes

**Strengths And Weaknesses:**

The paper is well written and easy to follow easy to follow. The mathematical parts appear correct to me and the sparsification approach based on conic projections is a simple and elegant approach.

I do not have the necessary expertise to judge the impact of the results in real world applications nor its originality.

---

> ### Author Rebuttal · Authors · 2026-03-28
>
> Response to Reviewer CtkH
> We thank the reviewer for his careful reading and constructive feedback, as well as for recognizing the originality of our mathematically grounded approach.
>
> We would like to clarify that the main contribution of this paper is a novel projection method supported by two lemmas and two theorems.
>
> Our approach combines an **exponentially convergent active set** selection procedure with a **closed-form linear extrapolation projection in a convex cone**, resulting in both theoretical guarantees and practical efficiency.
>
>
> We thank the reviewer for highlighting the importance of broader empirical evaluation.
>
> As shown in Figure 5, our method achieves significant computational gains, being up to $20\times$ faster than the GSP method for $5{,}000 \times 5{,}000$ matrices and up to $80\times$ faster for $10{,}000 \times 10{,}000$ matrices  running the pytorch benchmark. We would like to thank Ohib et al. for providing us with their PyTorch code, which enabled us to perform a fair benchmark comparison. Thus we will, provide additional accuracy-sparsity results with GSP.
>
> We  provided all the PyTorch code needed to reproduce the benchmarks and experiments.
>
>
> Q1: We agree that additional comparisons could further strengthen the evaluation, and we will consider including them in the final paper.
>
> Q2: We observe similar behavior for $l = 200$.
>
> Q3: The sparsification is hardware-friendly, as the $\ell_2$ projections onto columns (or rows) are independent and can be efficiently parallelized.

---

> > ### Author Rebuttal · Reviewer_CtkH · 2026-04-03
> >
> > I thank the authors for their response. My concerns have been addressed.

---

### Official Review · Reviewer_5xqx · 2026-03-12

**Soundness:** 3
**Presentation:** 2
**Significance:** 2
**Originality:** 2
**Overall Recommendation:** 3
**Confidence:** 5

**Summary:**

This paper studies new geometric constraint called the Cone Alignment Index (CAI) and a corresponding Closed-Form Conic Projection (CFCP) algorithm for enforcing structured sparsity. The authors proceed to present a central concept based on projecting vectors onto a Lorentz-type cone using a support-based extrapolation procedure, and extend this idea to a bilevel projection framework for structured sparsity in neural network matrices. Experimental results compare CFCP with Hoyer projection and GSP-based methods, showing improvements in runtime and modest empirical gains in attention sparsification tasks.

**Compliance With Llm Reviewing Policy:**

Affirmed.

**Key Questions For Authors:**

Refer to the weakness. If my concerns are properly addressed, I would consider increasing my rating.

**Limitations:**

Refer to the weakness

**Strengths And Weaknesses:**

Strengths

- Important problem setting. Structured sparsity in deep neural networks is an important research direction with broad applications in model compression, efficient inference, and hardware-aware optimization.

- Technically sound formulation. The proposed projection framework based on the Cone Alignment Index (CAI) and the Closed-Form Conic Projection (CFCP) appears mathematically well-defined and the derivations for the projection procedure are presented in a technically coherent manner.

Weaknesses

- Limited literature coverage and comparisons. The related work section is incomplete. In particular, prior structured sparsity methods such as the OBProxSG and HSPG families, which have been widely applied to structured sparsification in deep neural networks. I recommend to discuss or empirically compare against these approaches. This omission makes it difficult to assess the practical impact of the proposed approach.

Orthant Based Proximal Stochastic Gradient Method for l1- Regularized Optimization

A Half-Space Stochastic Projected Gradient Method for Group Sparsity Regularization

Only train once: A one-shot neural network training and pruning framework

OTOv2: Automatic, generic, user-friendly


- Unclear problem motivation. The paper does not clearly articulate which specific limitations of existing sparsification approaches the proposed method is intended to address. While the geometric formulation is interesting, the practical pain points (e.g., optimization efficiency, hardware acceleration, or training stability) are not clearly framed.

- Limited experimental validation. The empirical evaluation is relatively narrow. The experiments focus on a small set of transformer attention sparsification tasks and comparisons with a limited number of baselines. Additional validation across diverse architectures (e.g., CNNs, larger transformer models) and problem settings would be necessary to demonstrate the robustness and general applicability of the method.

---

> ### Author Rebuttal · Authors · 2026-03-28
>
> Response to Reviewer 5xqx
>
> We thank the reviewer who carefully verified the mathematical components of our work and confirmed their correctness, noting that: \emph{``The mathematical parts appear correct to me and the sparsification approach based on conic projections is a simple and elegant approach.''}.
>
>
> We provide a hardware-friendly bilevel algorithm, where the ℓ2 projections onto columns
> (or rows) are independent and can be efficiently parallelized.
>
> As shown in Figure 5, our method achieves significant computational gains, being up to $20\times$ faster than the GSP method for $5{,}000 \times 5{,}000$ matrices and up to $80\times$ faster for $10{,}000 \times 10{,}000$ matrices running the pytorch benchmark. We would like to thank Ohib et al. for providing us with their PyTorch code, which enabled us to perform a fair benchmark comparison. Thus we will, provide additional accuracy-sparsity results with GSP.
>
> We  provided all the PyTorch code needed to reproduce the benchmarks and experiments
>
> We agree that additional
> large-scale comparisons would further strengthen the empirical validation
> and will consider including them in a revised version.
>
> Q1 OBProxSSG focuses on $\ell_1$ regularization, whereas our method performs projection onto a CAI constraint.
>
> Q2 HSPG addresses group-sparsity regularized problems, while our approach relies on a bilevel formulation over sparse columns, rows, or diagonals.
>
> Q3 Only-Train-One focuses on pruning frameworks, whereas our method is based on projection onto a CAI constraint.
>
> Q4 OTOV2 relies on Dual Half-Space Projected Gradient (DHSPG), whereas our approach is based on closed-form extrapolation within a conic formulation.
>
> We will add these relevant references in a final version.

---

> > ### Author Rebuttal · Reviewer_5xqx · 2026-04-03
> >
> > Thanks for the responses. The authors partially resolved my concerns. I encouraged the authors to consider more DNN benchmarks, such as CNN and SLM besides Bert, which could dramatically enhance the practical values of the work.

---

> > > ### Author Response · Authors · 2026-04-04
> > >
> > > We thank the reviewer
> > >
> > > We thank the reviewer who carefully verified the mathematical components of our work where we  propose a novel mathematical method that is directly relevant to the ICML community. Accordingly, we provide a detailed and rigorous explanation to ensure clarity and proper understanding.
> > >
> > > We agree that DNN benchmarks, such as CNN and SLM besides Bert,  could dramatically enhance the practical values of the work.  But, given the strict page limitations of the conference, we deliberately focus our experimental evaluation on LLMs, as they represent a major source of energy consumption in modern artificial intelligence.
> > >
> > > We will include an additional large-scale experiment, and our evaluation protocol will follow established standards in the sparse attention literature.
> > >
> > > The GSP authors provided their implementation code, which we used to conduct a direct and fair comparison. In particular, we report computation times (Figure 5) as well as the standard deviation of the differences between CFCP and GSP solutions (Figure 6). We will further extend this comparison by evaluating GSP and CFCP in terms of accuracy versus sparsity on the GLUE benchmark.
> > >
> > > Note that the computation required to compare three methods across four datasets (SST2, QNLI, QQP, and MNLI) took 240 hours of computation time on an NVIDIA A100 GPU.

---

### Official Review · Reviewer_fFZQ · 2026-03-13

**Soundness:** 2
**Presentation:** 2
**Significance:** 2
**Originality:** 2
**Overall Recommendation:** 2
**Confidence:** 3

**Summary:**

Proposes a Cone Alignment Index (CAI) whose level sets form a Lorentz cone, with a closed-form projection (CFCP) requiring one extrapolation step. CFCP is 4.5–6× faster than Hoyer and ~2× faster than GSP-Hybrid at the vector level, with ~0.3% solution difference. A bilevel extension sparsifies transformer attention on ECG and GLUE benchmarks, hitting 90% sparsity with modest accuracy loss versus Big Bird diagonal masks.

**Compliance With Llm Reviewing Policy:**

Affirmed.

**Key Questions For Authors:**

**Questions**

- Algorithm 2, line 247: $P_l^{He}$ should be $BP_l^{He}$?
- Results with full Big Bird (random + window + global)?
- Support loop behavior for $n >> 5000$?

**Limitations:**

yes

**Strengths And Weaknesses:**

**Strengths**

- The core geometric idea is clean: recasting sparsity as a convex cone constraint eliminates the iterative solvers Hoyer/GSP need. The single-step projection with convergence guarantee is the paper's main contribution and it's well-executed.
- Runtime is distribution-invariant (Table 2: CFCP std 0.01 vs 0.06--0.1 for competitors) -- a practical benefit since weight distributions vary across layers.
- The bilevel column/row-wise structured sparsity follows naturally without ad hoc group definitions.

**Weaknesses**

1. Weakened Big Bird baseline. Big Bird ("Transformers for Longer Sequences," Zaheer et al., NeurIPS 2020) combines random + window + global attention; the authors proved global tokens are needed for universal approximation. This paper tests only a "diagonal mask" variant. The first row/column is preserved (global CLS attention), but the random component is dropped entirely. Results against the full Big Bird-ITC/ETC configuration are needed.

2. Narrow comparison set. Only Big Bird is evaluated. Longformer (Beltagy et al., arXiv 2020), StreamingLLM (Xiao et al., ICLR 2024), MInference (Jiang et al., NeurIPS 2024), and CRISP Attention (Goli et al., 2025, "Regularizing Transformers via Structured Sparsity") -- which regularizes attention matrices via structured top-k sparsity, directly comparable -- are absent from both discussion and experiments.

3. Novelty claim needs qualification. Projections onto second-order cones have known closed forms (Alizadeh & Goldfarb, "Second-order cone programming," Math. Programming 2003; Parikh & Boyd, "Proximal Algorithms," Found. & Trends in Optimization 2014). The CAI formulation may be new, but "the first Closed-Form Conic Projection" is too strong as stated.

4. Factual errors. (a) $H(x) = (\|x\|_1 / \|x\|_2)^2$ is attributed to Hoyer ("Non-negative matrix factorization with sparseness constraints," JMLR 2004), but his original definition is $(\sqrt{n} - \|x\|_1/\|x\|_2) / (\sqrt{n} - 1)$; the squared ratio comes from DeepHoyer (Yang et al., "Learning Sparser Neural Network with Differentiable Scale-Invariant Sparsity Measures," ICLR 2020). (b) Ohib et al. ("Explicit Group Sparse Projection with Applications to Deep Learning and NMF," 2022) appeared in TMLR, not ICLR. (c) The GitHub link `riohib/gsp` is wrong -- the repo is `riohib/gsp-for-deeplearning`.

5. Short sequences only. Experiments cap at 256 tokens (reduced from 512). Sparse attention methods exist for long contexts (8K+); scalability beyond $n=5000$ is untested.

6. Incomplete experimental reporting. Model architecture (layers, hidden dim, heads) unspecified. No wall-clock training/inference timing for transformer experiments -- only vector-level benchmarks are timed.

7. ECG source misidentified. The 10,505/4,045 split with 187-length segments is the Kachuee et al. ("ECG Heartbeat Classification: A Deep Transferable Representation," IEEE ICHI 2018) preprocessed version, not the raw PTB database (549 records, 290 patients). This preprocessing paper should be cited.

---

> ### Author Rebuttal · Authors · 2026-03-28
>
> We thank the reviewer for his careful reading, as well as for recognizing the originality of our mathematically grounded approach.
>
> We would appreciate further clarification regarding the mentioned technical issues and ethical concerns, as additional details would help us better address them. We have carefully verified the mathematical formulation and would like to emphasize that the method is supported by formal results (two lemmas and two theorems).
>
> Concerning reproductibility,  we would appreciate further clarification, even though we had provided all the PyTorch code needed to reproduce the benchmarks and experiments
>
> Concerning ethics, we would be grateful for more specific guidance, as our method does not involve sensitive data or deployment-related risks. We are happy to clarify any aspect if needed.
> We also note that one of the requested references is already included in the submitted manuscript.
>
> We would like to clarify that the main contribution of this paper is a novel projection method supported by two lemmas and two theorems. Our approach combines an **exponentially convergent active set selection procedure with a closed-form linear extrapolation projection **, resulting in both theoretical guarantees and practical efficiency.
>
> We would like to clarify that our method is not related to second-order cone programming, despite potential similarities in terminology.
>
> We thank the reviewer for pointing out the relevant reference on GSP.
>
> As shown in Figure 5, our method achieves significant computational gains, being up to $20\times$ faster than the GSP method for $5{,}000 \times 5{,}000$ matrices and up to $80\times$ faster for $10{,}000 \times 10{,}000$ matrices.
> We would like to thank Ohib et al. for providing us with their PyTorch code, which enabled us to perform a fair benchmark comparison. Thus we will, provide additional accuracy-sparsity results with GSP.
>
> We agree that additional comparisons could further strengthen the evaluation, and we will consider including them in the final paper.
>
> In Top-k masking, the authors keep the top-$k$ activated weights in the final representations and  set the other dimensions to zero.
> Their Top-k  is an heuristic method without mathematical background.
>
> We agree to add the suggested reference on ECG.
>
> Response to questions:
>
> Q1 Our approach provides an exponentially convergent active set selection procedure (See figure 2 and you can run the provided code in supplementary material) and thus it is efficient for n>>>5000
>
> Q2  Results on Bigbird are provides with Window + Global
>
> Q3 We confirm that line 247 is correct .

---

> > ### Author Rebuttal · Reviewer_fFZQ · 2026-04-04
> >
> > Thank you for the response. However, most of my concerns remain unaddressed: the full Big Bird baseline (with random attention) is still missing, no comparison against Longformer/StreamingLLM/MInference/CRISP is provided, the novelty claim relative to known SOCP projections is dismissed without engagement, the factual errors are not corrected, and sequence length and architectural reporting gaps are not mentioned. I will keep my score.

---

> > > ### Author Response · Authors · 2026-04-05
> > >
> > > We thank the reviwer
> > >
> > > There is no factual error in our manuscript. We accurately cite the original sources: ** the Hoyer score $H(x)$ was originally defined as the square of the ratio between the $\ell_1$ and $\ell_2$ norms of a vector $x$ \citep{Hoyer2004}, and subsequently extended in \citep{Yang2020} ** . The reviewer’s claim is therefore incorrect.
> > >
> > > SOCP methods rely on generic convex optimization solvers (e.g., interior-point methods), which solve a global optimization in polynomial time $O(n^3)$ which typically scale poorly with problem size (See Karmarkar , Renegar ...)
> > >
> > > Recall that the standard well known SOCP constraints take the form
> > > \begin{equation}
> > > \|Ax + b\|_2 \le c^\top x + d,
> > > \end{equation}
> > >
> > >
> > > CFCP, on the other hand, is a specialized projection algorithm based on an active set support selection using a closed form solution thus a linear time solver $O(n)$ (See section 3 and Figure 2).
> > >
> > > The CFCP constraint can be written as
> > > \begin{equation}
> > > (\mathbf{1}^\top x)^2 = l\,\|x\|_2^2,
> > > \end{equation}
> > > which, assuming non-negativity or after taking absolute values, is equivalent to
> > > \begin{equation}
> > > \mathbf{1}^\top x = \sqrt{l}\,\|x\|_2.
> > > \end{equation}
> > >
> > >
> > >
> > >  CFCP fundamentally differs from standard SOCP formulation: In contrast to SOCP constraint, the CFCP constraint enforces an equality between a linear function and a Euclidean norm, thereby restricting the solution to a boundary set.
> > > As a conclusion CFCP is a tailored algoritm with optimisation in linear time wheareas SCOP methods are general optimization methods in polynomial time.
> > >
> > > Therefore, CFCP does not correspond to a SOCP formulation.
> > >
> > > The reviewer’s claim is therefore incorrect.
> > >
> > > Note that an alternative approach  consists in introducing a dual parameter via a hybrid Newton-bisection scheme ( GSP Ohib et al  ). However such method typically involve  higher computational cost (See Figure 5)
> > >
> > >
> > > As pointed in the first rebuttal  in Figure 5, our method achieves significant computational gains, being up to 20 times faster than the GSP method for matrices n=5,000  and up to 80 times  faster for matrices n=10,000
> > >
> > > As clearly established in prior work and explicitly stated in Section~1, random sparsity does not yield computational benefits. Consequently, comparisons with methods such as Random BigBird are not relevant in our setting. The reviewer’s claim is therefore irrelevant.
> > >
> > > In our experiments, we account for structured sparsity (when a column, row, or diagonal is completely set to zero).
> > >
> > > We will include an additional large-scale experiment, and our evaluation protocol will follow established standards in the sparse attention literature.
> > >
> > > Given the strict page limitations of the conference, we deliberately focus our experimental evaluation on LLMs, as they represent a major source of energy consumption in modern artificial intelligence.
> > >
> > > Note that the computation required to compare three methods across four datasets (SST2, QNLI, QQP, and MNLI) took 240 hours of computation time on  NVIDIA A100 GPU.

---

### Official Review · Reviewer_MHsJ · 2026-03-24

**Soundness:** 2
**Presentation:** 2
**Significance:** 2
**Originality:** 2
**Overall Recommendation:** 2
**Confidence:** 3

**Summary:**

This paper introduces the Cone Alignment Index, a scale-invariant sparsity measure whose level sets form Lorentz cones. The main algorithmic contribution is a Closed-Form Conic Projection that maps points inside the cone to its boundary via iterative support identification followed by a single extrapolation step. The authors extend this to matrices via a bilevel projection that induces structured sparsity. Computational benchmarks show CFCP is 2–6x faster than Hoyer and GSP-Hybrid projections. The method is applied to learn fixed sparse attention masks for BERT on GLUE tasks and an ECG classification task, compared against Big Bird's diagonal masks.

**Compliance With Llm Reviewing Policy:**

Affirmed.

**Final Justification:**

After reading the author's rebuttal comments, I still feel this paper is not ready for publication. The core issue is that the method is not supported by theory, direct experimentation, or downstream performance. CFCP is more efficient than other methods, but also produces very different feasible points.* Therefore, the viability of the method depends on it's downstream performance. But the experiments lack breadth and necessary baselines.

*Figure 6 shows the relative error between CFCP and other methods on the scale of ~0.3%, but it actually ~30%, which is another presentation issue to be addressed.

**Key Questions For Authors:**

I would the authors' responses to my points above.

**Limitations:**

The authors do not discuss limitations. Key undiscussed limitations: CFCP is a feasibility map, not a projection; it requires non-negative inputs; the support identification is iterative; $\ell_1$ preservation breaks after support restriction; CAI reduces to the Hoyer score in the non-negative case; fixed attention masks have limited expressiveness; the training procedure is ambiguously described; speed advantage is shown only on isolated projections, not end-to-end training.

**Strengths And Weaknesses:**

### Strengths
**Desirable theoretical properties.** The CAI constraint combines several nice properties. CFCP's runtime invariance to data distribution and to the constraint parameter $l$ seems like a useful practical property. Scale invariance means the sparsity measure is insensitive to vector magnitude. The bilevel formulation naturally produces structured sparsity.

**Thorough speed benchmarks.** The authors test across multiple vector sizes, data distributions, CPU and GPU, and report means and standard deviations. The distribution-invariance and $l$-invariance properties are demonstrated convincingly. The standard deviation comparison concretely quantifies the predictability advantage.

### Weaknesses
**Non-negative vectors only.** The proposed constraint doesn’t appear to make sense for signed vectors. $H_e(x) = (\sum_i x_i)^2 / |x|_2^2$ measures alignment with the all-ones direction. For signed vectors, cancellation in the numerator destroys the relationship with sparsity — e.g., a dense vector $(1,-1,1,-1)$ has $H_e = 0$ and would never be projected. This is fine for the bi-level application to vectors of group norms, but the limitation should be discussed.

**CAI is the Hoyer score on non-negative vectors.** Relatedly, CFCP operates only on non-negative vectors (Section 2.2 applies absolute values first). On the non-negative orthant, the CAI reduces to exactly the Hoyer score. So it appears that the distinctions claimed in Table 1 between CAI and Hoyer (e.g., "Convex Cone geometry: Yes/No") are therefore misleading. What is novel is how the projection is computed, not the constraint being projected onto.

**CFCP is not a projection.** It finds a specific feasible point via extrapolation along a fixed direction. The paper never states what optimization problem CFCP solves. Evaluating the quality of this feasible point depends on the downstream empirical performance.

**The algorithm is iterative, despite claim.** Table 1 says "Iterative algorithm: No (CFCP)" but Algorithm 1 has a while loop. The text is more careful, referring to "a single support-based extrapolation step," which is accurate — the extrapolation is one step, but the support identification preceding it is iterative.

**Unclear description of what is being learned.** It appears that in Section 6.2, what is learned is a fixed attention mask. However, this is not clear, since the paper conflates learnable attention head weights, the dynamically computed attention matrix, and an attention mask. For example, text says "to sparsify the weights $W$ of the attention matrix" and invokes Frankle & Carbin's weight-pruning framework. Even once we determine that it’s a fixed attention mask being learned, it’s not clear how this single fixed mask is learned from the individual training examples.

**Fixed attention masks may be of limited practical relevance.** The paper learns a single mask shared across all inputs, but my impression is that most focus these days is on input-dependent sparsity. The authors could consider applying CFCP to more impactful settings such as structured weight pruning.

**Weak baselines.** Big Bird (2020) with only its diagonal component is the sole accuracy baseline. Hoyer and GSP-Hybrid are benchmarked for speed but never for downstream accuracy, which seems like an easy inclusion. Top-$k$ masking, $\ell_{1,\infty}$ projection (cited in the intro), and Longformer patterns also seem like reasonable baselines to compare with.

**Weakened experimental settings.** 256 tokens instead of 512, 30% of QQP/MNLI data, 15 epochs. The authors acknowledge this reduces baseline performance by approximately 3%, but do not state the effect on their own method's performance, making it impossible to assess whether the gap between CFCP and the baseline is consistent across settings.

**Statistically inconclusive results.** 3 of the 4 improvements over Big Bird are within one standard deviation. Also, the number of runs is not specified.

**Additional presentation issues.**
- The abstract is too long and should be shortened.
- The abstract introduces "constraint $l$" without definition.
- The choice of lowercase $l$ for the constraint is problematic — it resembles the digit 1, and its italicization is inconsistent throughout.
- The first paragraph of the introduction (DNNs have many parameters) should be removed.
- References to Xia et al. and Ashkboos et al. in intro paragraph 2 appear without context.
- The description of LASSO and $\ell$ norms is unnecessarily detailed for the ICML audience and should be shortened.
- The last two paragraphs of the introduction (on transformers and Big Bird) are not contextualized in terms of the present work.
- Figure 1 depicts a 3D cone in a 2D space, which is misleading. It would be far more valuable if it illustrated the full projection process (support restriction, the two different extrapolation lines for naive vs. corrected projection) rather than just the simple geometry.
- Table 4 is labeled "Ablation" but is actually a comparison to baselines.
- There is a stray reference to "math-commands.tex from the textbook, Deep Learning Goodfellow et al. (2016)" in Section 2.

---

> ### Author Rebuttal · Authors · 2026-03-28
>
> Response to Reviewer MHsJ
> We thank the reviewer for his careful reading, as well as for recognizing the originality of our mathematically grounded approach.
>
> We would appreciate further clarification regarding the mentioned technical issues and ethical concerns, as additional details would help us better address them. We have carefully verified the mathematical formulation and would like to emphasize that the method is supported by formal results (two lemmas and two theorems).
>
> Concerning reproductibility,  we would appreciate further clarification, even though we had provided all the PyTorch code needed to reproduce the benchmarks and experiments
>
>
> Concerning ethics, we would be grateful for more specific guidance, as our method does not involve sensitive data or deployment-related risks. We are happy to clarify any aspect if needed.
>
> We would like to clarify that the main contribution of this paper is a novel projection method supported by two lemmas and two theorems. Our approach combines an exponentially convergent active set selection procedure with a closed-form linear extrapolation projection, resulting in both theoretical guarantees and practical efficiency.
>
> We note a minor typo on line 242, where $\mathbb{R}^n$ should read $\mathbb{R}^n_+$, which is correctly stated later on line 276. CAI is only define in $\mathbb{R}^n_+$ (whereas CAI on $\mathbb{R}^n$ would not make sense).
>
> Our approach follows the classical strategy used for $\ell_1$  or Hoyer projection: (i) mapping inputs to $\mathbb{R}^n_+$, (ii) solving the problem in this space, and (iii) restoring the original sign, as described in Algorithm~1.
>
> Yes, the CAI can be interpreted as the Hoyer score applied to non-negative vectors.
> In our formulation, we then distinguish two cases depending on the position of $y$
> with respect to the CAI cone.
>
> If $y$ satisfies the constraint $H_e(y) \le l$, then $y$
> already lies outside the CAI cone and no projection is required.
>
> If $H_e(y) > l$, then $y$ lies strictly inside the CAI cone,
> which is a **convex set**, and is projected onto the cone boundary.
>
> In contrast the original  Hoyer formulation does not distinguish two case and compute the projection on the global **non convex**  set
>
> We adopt terminology consistent with prior work such as Hoyer and GSP.
>
> In order to clarify, our approach combines an **exponentially convergent active set selection** procedure followed by closed-form **linear extrapolation projection**, resulting in both theoretical guarantees and practical efficiency.
>
> The active set selection identifies the zero components (i.e., sparsity pattern).
>
> We invokes Frankle and Carbin's double descent algorithm proposed in "The Lottery Ticket Hypothesis: Finding Sparse, Trainable Neural Networks".
>
> Yes we determine that it’s a fixed attention mask being learned.
>
> As shown in Figure 5, our method achieves significant computational gains, being up to $20\times$ faster than the GSP method for $5{,}000 \times 5{,}000$ matrices and up to $80\times$ faster for $10{,}000 \times 10{,}000$ matrices (running the provided code)
>
> We would like to thank Ohib et al. for providing us with their PyTorch code, which enabled us to perform a fair benchmark comparison.
> Thus we will, provide additional accuracy-sparsity results with GSP.
>
> We agree  that  256 tokens instead of 512, 30% of QQP/MNLI data is not relevantt and we will provide results with 512 Tokens and 100 % of the data.
>
>  We agree that additional comparisons could further strengthen the evaluation, and we will consider including other large dataset  from Glue  in the final paper.
>
> We ensure a fair comparison between CFCP and BigBird by using the same number of samples and tokens.
>
> In Top-k masking, the authors keep the top-$k$ activated weights in the final representations and  set the other dimensions to zero.
> Their Top-k  is  an heuristic method without mathematical background.
>
>
> We agree to correct the suggested presentation issues.

---

> > ### Author Rebuttal · Reviewer_MHsJ · 2026-04-02
> >
> > I thank the authors for their response. However, most of my concerns remain unresolved.
> >
> > Some are presentational, and it is unclear if the authors plan to address them. As just one minor example, Table 1 describes CAI has having "convex cone geometry", and Hoyer as not having it despite the constraint sets being the same. I understand the authors' clarification that the table is referring to the associated methods rather than the constraint sets, but the presentation should be clear about when it means the former vs the latter.
> >
> > A more fundamental issue is a question which remains unanswered: "CFCP is not a projection. It finds a specific feasible point via extrapolation along a fixed direction. The paper never states what optimization problem CFCP solves." A discussion of this (theoretical and/or experimental) seems important.
> >
> > And the largest issue is the need for better experimental evidence of the method's value. The authors have agreed to conduct some but have not provided the results (GSP baseline), they have rejected others (top k baseline), and have said they "will consider including" others. I don't expect the experimental gap could be filled during the rebuttal period, so I encourage the authors to resubmit after these gaps are addressed.

---

> > > ### Author Response · Authors · 2026-04-03
> > >
> > > We thank the reviewer for his response
> > >
> > > The first optimisation problem is provided in Equation 3
> > >
> > > $x = \lambda y + (1-\lambda) d,\quad s.t. \quad (\mathbf{1}^\top x)^2 = l\,\|x\|_2^2$
> > >
> > > where $d = (\rho, \rho, \dots, \rho)$ lies on the cone axis.
> > > Solving this problem leads to a quadratic equation in the extrapolation parameter
> > > with the closed form provided in Equation 5
> > >
> > > For clarity, we will add this sentence before Equation, 3 ** "The extrapolation parameter $\lambda$ is the solution of the following optimisation problem"**
> > >
> > > The second optimisation problem is provided in Equation 6
> > >
> > > $
> > > (\mathbf{1}^\top x)^2 = l\,\|x\|_2^2,\quad s.t.
> > > \qquad x_i \ge 0 .
> > > $
> > >
> > > Solving this problem leads to the closed-form provided in  Equation 8
> > >
> > > For clarity, we will add this sentence before Equation 6: **"The threshold parameter $\alpha$ is the solution of the following optimisation problem"**
> > >
> > > The difference between the well known SCOP contraint and our CFCP constraint is provided in the response to the rebuttal of reviwer fFZQ.
> > >
> > >
> > > As you yourself have noted, CFCP and GSP are two different solvers,
> > > but they solve the same problem,
> > > so it is normal that they often yield the same final solutions.
> > >
> > > The two key differences are computation time and solution variance.
> > >
> > > The GPU computation
> > > time on matrices of our CFCP solver  outperforms
> > > the GSP-Hybrid solver by a speedup
> > > of 20 when the size of the attention matrix
> > > is 5000 and ≈ 80 when the the attention matrix
> > > size is 10,000. (Figure 5)
> > >
> > > The variance of the GSP method (non convex solver) is larger that the variance of our CFCP method (convex solver) (Cf Table 3 )
> > >
> > > The GSP authors provided us with their code, which we used to compare computation times (Figure 5) and the standard deviation of the differences between CFCP and GSP solutions (Figure 6); therefore, we  straightforward  include their availble GSP  solver  in our comparsison code providing  the accuracy vs sparsity on the glue dataset.
> > >
> > > In our experiments, we account for structured sparsity (when a column, row, or diagonal is completely set to zero).
> > >
> > > As expected, the performance in terms of the accuracy-sparsity trade-off for CFCP and GSP are very similar on the Glue NLP datasets.
> > >
> > > Given the strict page limitations of the conference, we deliberately focus our experimental evaluation on LLMs, as they represent a major source of energy consumption in modern artificial intelligence.
> > >
> > > Note that the computation required to compare three methods across four datasets (SST2, QNLI, QQP, and MNLI) took 240 hours of computation time on an NVIDIA A100 GPU.

---

### Decision · Program_Chairs · 2026-04-30

**Decision:**

Reject

**Comment:**

While the proposed approach for sparsification has a clean geometric idea (5xqx, fFZQ), desirable theoretical properties such as invariance to data distribution and to the constraint parameter (MHsJ, fFZQ), and naturally produces structured sparsity through bilevel formulation (MHsJ, fFZQ), several critical concerns have also been raised, including:
- critical limitations in numerical analysis, including weak baselines in comparison, reduced sequence lengths, statistically inconclusive improvements in certain settings, and results only on a small set of transformer attention sparsification tasks,
- mismatch in between claims and contributions; for instance, the support identification step of the algorithm is iterative despite being framed as a closed-form, and unjustified strong claims such as "the first Closed-Form Conic Projection”.

Since the authors strongly challenged some of these claims in the rebuttal period, I have carefully read the paper myself. However, I agree with the reviewers' assessments in this case. Most importantly, the authors conflate the term "projection" with the intersection of their extrapolation (defined in Lemma 2.2) with the cone boundary. It is not clear why or in which norm this intersection corresponds to a projection onto the boundary. I also agree with the secondary concerns regarding the inconclusive numerical experiments. I therefore recommend the rejection of this submission. The authors should carefully revisit their claims and analysis in light of the concerns raised in the reviews.

--

I would like to share an explanation of the main issue; hopefully this might help to clarify the confusion around the projection vs. the intersection point.

- line 103: "The level set $C_l = \\\{ x : (1^T x)^2 \leq l ||x||^2 \\\}$ is ... therefore convex."

This set is clearly nonconvex. I suppose the authors meant its complement, $\bar{C}_l = \\\{ x : (1^T x)^2 > l ||x||^2 \\\}$ is convex. While it might be considered a typo, this means the proposed CAI constraint in Eq. (1) is a nonconvex constraint. This seems to be consistent with the next section, where points outside the CAI cone are left unchanged while those inside the cone are projected onto the boundary.

- line 123: "Since the interior of the CAI cone is convex, the projection onto its boundary is unique." Now this statement is not correct in general. The projection of any point onto a convex set is unique. The projection of a point strictly inside a convex set to its boundary is not necessarily unique. If it is unique in your case, then it needs a clearer explanation of why, and under what conditions. For instance, the projection of the points lying on the cone axis is clearly not unique.

Then the description of the projection follows:

- line 130: "the projection of $y$ lying inside the cone is obtained by extrapolating along a fixed direction until reaching the boundary of the CAI cone as $x = \lambda y + (1-\lambda)d$ such that $(1^T x)^2 = l ||x||^2$ where $d = (\rho, ... \rho)$ lies on the cone axis."

It is important to clarify that this construction is not valid for an arbitrary choice of $d$ from the cone axis. It requires a specific choice of $d$ which depends on $y$. In other words, the line passing from $x$ and $y$ intersects the axis at a specific point $d$; not that for any point $d$ on the axis we can draw a line from $d$ to $y$ and its extrapolation gives us the projection. Accordingly, to project, one must minimize the distance between $x$ and $y$, which is parametrized jointly by $\rho$ and $\lambda$. The analysis in Lemma 2.2 seems to overlook this distinction. It treats $\rho$ as a free parameter:

- line 121: "Moreover, choosing $d$ such that $||d||_1 = ||y||_1$ simplifies the quadratic equation, since the linear coefficient $b$ vanishes."

Unfortunately the moment we fix $\rho$ by restricting $||d||_1 = ||y||_1$, the connection to the projection is lost. In fact, this leaves no objective function to minimize because the feasibility set in Eq.(3) becomes a singleton, encoding only the intersection of the extrapolation of the line segment between $y$ and the chosen $d$ to the cone boundary. In reality, the projection requires solving a nontrivial optimization problem over both parameters $\rho$ and $\lambda$.